

# Top-down constraints on global N₂O emissions at optimal resolution: application of a new dimension reduction technique

Kelley C. Wells[1], Dylan B. Millet[1], Nicolas Bousserez[2], Daven K. Henze[2], Timothy J. Griffis[1], Sreelekha Chaliyakunnel[1], Edward J. Dlugokencky[3], Eri Saikawa[4], Gao Xiang[5], Ronald G. Prinn[6], Simon O'Doherty[7], Dickon Young[7], Ray F. Weiss[8], Geoff S. Dutton[3,9], James W. Elkins[3], Paul B. Krummel[10], Ray Langenfelds[10], and L. Paul Steele[10]

[1]Department of Soil, Water, and Climate, University of Minnesota, St. Paul, Minnesota, USA
[2]Department of Mechanical Engineering, University of Colorado at Boulder, Boulder, Colorado, USA
[3]Earth System Research Laboratory, NOAA, Boulder, Colorado, USA
[4]Department of Environmental Sciences, Emory University, Atlanta, Georgia, USA
[5]Joint Program on the Science and Policy of Global Change, Massachusetts Institute of Technology, Cambridge, Massachusetts, USA
[6]Center for Global Change Science, Massachusetts Institute of Technology, Cambridge, Massachusetts, USA
[7]School of Chemistry, University of Bristol, Bristol, UK
[8]Scripps Institute of Oceanography, University of California San Diego, La Jolla, California, USA
[9]CIRES, University of Colorado at Boulder, Boulder, Colorado, USA
[10]Climate Science Centre, CSIRO Oceans and Atmosphere, Aspendale, Victoria, Australia

*Correspondence to*: Dylan B. Millet (dbm@umn.edu)

**Abstract.** We present top-down constraints on global, monthly N₂O emissions for 2011 from a multi-inversion approach and an ensemble of surface observations. The inversions employ the GEOS-Chem adjoint and an array of aggregation strategies to test how well current observations can constrain the spatial distribution of global N₂O emissions. The strategies include: (1) a standard 4D-Var inversion at native model resolution (4° × 5°), (2) an inversion for six continental and three ocean regions, and (3) a fast 4D-Var inversion based on a novel dimension reduction technique employing randomized singular value decomposition (SVD). The optimized global flux ranges from 15.9 Tg N yr⁻¹ (SVD-based inversion) to 17.5-17.7 Tg N yr⁻¹ (continental-scale, standard 4D-Var inversions), with the former better capturing the N₂O background measured during the HIAPER Pole-to-Pole Observations (HIPPO) airborne campaigns. We find that the tropics provide a greater contribution to the global N₂O flux than is predicted by the prior bottom-up inventories, likely due to underestimated agricultural and oceanic emissions. We infer an overestimate of natural soil emissions in the extratropics, and find that predicted emissions are seasonally biased in northern midlatitudes. Here, optimized fluxes exhibit a springtime versus summertime peak more consistent with the timing of fertilizer application, soil thawing, and elevated soil moisture. Finally, the inversions reveal a major emission underestimate in the US Corn Belt (which may extend to other intensive agricultural regions), likely from underrepresentation of indirect N₂O emissions from leaching and runoff. We extensively test the impact of initial conditions on the analysis and recommend formally optimizing the initial N₂O distribution to avoid aliasing the inferred fluxes. We find that the SVD-based approach provides a powerful framework for deriving emission information from N₂O observations: by





defining the state vector based on the information content of the inversion, it provides useful spatial information that is lost when aggregating to ad-hoc regions, while also better resolving temporal features than a standard 4D-Var inversion.

## 1 Introduction

Nitrous oxide is a long-lived greenhouse gas ($\tau$ ~122-131 years; Volk et al., 1997; Prather et al., 2012) with substantial
impacts on both climate and stratospheric chemistry. It has a global warming potential far exceeding that of $CO_2$ ($265\times$ on a 100-year timescale; Myhre et al., 2013), and its emissions weighted by ozone depletion potential currently exceed those of all other substances (Ravishankara et al., 2009). The global $N_2O$ source is reasonably well constrained (15.7 to 20.1 Tg N yr$^{-1}$ for years 1999-2009; Prather et al., 2012; Saikawa et al., 2014; Thompson et al., 2014a; Thompson et al., 2014c) by its atmospheric abundance and estimated lifetime. However, attribution of the source to specific regions and sectors is hindered
by the strong spatio-temporal variability in $N_2O$ emissions combined with weak variability in atmospheric mixing ratios and a sparse global observing network (e.g., Wells et al., 2015). Quantitative interpretation of atmospheric $N_2O$ measurements in terms of globally resolved emissions thus first requires a rigorous assessment of how results hinge on the modeling framework employed. Here, we apply a hierarchy of model resolutions, including a new method that formally defines the state vector for optimization based on the information content of the observations, in a global inverse modeling framework to
address this need. We use this model hierarchy with a global suite of observations to: i) quantify the spatial and seasonal distribution of $N_2O$ emissions for 2011, ii) examine what features of these results are robust across model configurations, and iii) assess the implications for current understanding of the $N_2O$ budget and future research needs.

The primary sources of atmospheric $N_2O$ are microbial denitrification and nitrification, which lead to $N_2O$ production in soils (Firestone and Davidson, 1989), ocean waters (Elkins et al., 1978; Cohen and Gordon, 1979), and in streams, rivers,
and lakes (Seitzinger and Kroeze, 1998; Beaulieu et al., 2011). Global mean $N_2O$ mixing ratios rose by $0.85 \pm 0.1$ ppb yr$^{-1}$ from 2001-2015 (based on NOAA surface measurements) primarily due to increased use of nitrogen-based fertilizers (Galloway et al., 2008; Davidson, 2009; Park et al., 2012) and the nonlinear response of $N_2O$ emissions to fertilizer application in some agricultural systems (Shcherbak et al., 2014). Bottom-up estimates for the global agricultural flux range from 4.3-6.3 Tg N yr$^{-1}$ (Mosier et al., 1998; Crutzen et al., 2008; Davidson, 2009), which include emissions occurring on-
field (i.e. 'direct' emissions from fertilized crops), downstream (so-called 'indirect' emissions from leaching and runoff), and from manure management. However, a body of recent work suggests that indirect $N_2O$ emissions downstream from the site of fertilizer application could be 2.6-9 times larger than is presently accounted for in bottom-up estimates (Griffis et al., 2013; Turner et al., 2015), which would imply an underestimate of the importance of agricultural sources for the overall $N_2O$ budget. Non-agricultural soils and oceans are thought to contribute an additional 7.4-11 Tg N yr$^{-1}$ (Saikawa et al., 2013) and
1.2-6.8 Tg N yr$^{-1}$ (Nevison et al., 1995; Jin and Gruber, 2003; Manizza et al., 2012), respectively, to the global $N_2O$ source. Industrial, transportation, and biomass burning emissions also exist but are thought to be relatively minor, totaling 1.2-1.8 Tg N yr$^{-1}$ (Prather et al., 2001).





Because microbial nitrification and denitrification in soils depend strongly on soil moisture, temperature, soil type, and fertilizer application timing (e.g., Potter et al., 1996; Bouwman, 1998; Bouwman et al., 2013; Butterbach-Bahl et al., 2013), $N_2O$ emissions can exhibit major temporal and spatial variability. For example, Wagner-Riddle et al. (2017) estimated that short-duration freeze-thaw cycles account for 35-65% of the total annual $N_2O$ emissions from seasonally frozen croplands

globally. This type of variability poses a major challenge to bottom-up and top-down efforts to quantify $N_2O$ surface fluxes and attribute them to specific times, locations, and mechanisms. The relatively sparse coverage of measurement sites and low atmospheric variability (surface mixing ratios typically vary by < 10 ppb due to the long $N_2O$ lifetime) compound the challenge, and limit the spatial and temporal resolution at which emission fluxes can be inferred (Wells et al., 2015). As a result, global $N_2O$ inversions often employ some aggregation strategy to optimize emissions for a small set of geographic

regions (e.g., Hirsch et al., 2006; Huang et al., 2008; Saikawa et al., 2014). However, this aggregation is typically done in an informal, ad-hoc way rather than by formally determining the degrees of freedom (DOFs) in the inverse system – which leads to aggregation errors and sub-optimal results. Work on $CO_2$ inversions has also highlighted this issue (e.g., Kaminski et al., 2001) and the resulting importance of determining the proper state vector size for optimal results (Bocquet et al., 2011).

Another key challenge is that because of the long $N_2O$ lifetime, inaccuracies in model initial conditions can lead to large

biases in the subsequent optimized emissions (Thompson et al., 2014c). Past global $N_2O$ inversion studies have established the initial conditions in a variety of ways: from a forward model spin-up that is then evaluated against observations (e.g., Huang et al., 2008); by including the initial condition as a separate adjustable parameter in the source optimization (e.g., Saikawa et al., 2014; Thompson et al., 2014a); or from interpolation of atmospheric observations (e.g., Wells et al., 2015). To our knowledge there has not yet been a detailed evaluation of these different methods and their impacts on $N_2O$ source

inversions. Such information is needed to establish the degree to which uncertainties in the initial conditions can propagate to errors in the optimized $N_2O$ emission estimates.

In this paper, we address the above uncertainties in a quantitative way using a multi-inversion hierarchy to derive top-down constraints on $N_2O$ emissions for 2011. We use the adjoint of the GEOS-Chem chemical transport model (CTM) to solve for monthly fluxes at the model grid box scale as well as at geographically aggregated continental scales. We compare these

results with those obtained using a new dimension reduction technique based on the SVD of the so-called prior-preconditioned Hessian of the 4D-Var cost function (Bousserez and Henze, 2017). This new SVD-based approach allows us to solve for fluxes at optimal spatiotemporal scales, as defined by the information content of the $N_2O$ observations – thus maximizing the DOFs for the inversion and avoiding any need for spatial aggregation based on geography or source type. It also offers significant time savings over standard grid-based 4D-Var approaches, due to the use of efficient randomized SVD

algorithms (Halko et al., 2011). The initial conditions for the above inversions are constructed in a variety of ways, and we use observations and model simulations to assess their accuracy and associated impacts on optimized $N_2O$ fluxes. We then evaluate these optimized emissions using airborne measurements and interpret the results in terms of underlying emission processes, with specific emphasis on the role of model resolution in affecting the solution, and on those features that appear most robust (and most uncertain) across model configurations.



## 2 Methods

### 2.1 GEOS-Chem N₂O simulation

The N$_2$O simulation employed here, previously described by Wells et al. (2015), is based on the GEOS-Chem CTM (www.geos-chem.org) with GEOS-5 assimilated meteorological data from the NASA Goddard Earth Observing System. We

use a horizontal resolution of 4° × 5° with 47 vertical levels from the surface to 0.01 hPa, and time steps of 30 minutes for transport and 60 minutes for emissions and chemistry. The simulation period spans April 2010-April 2012.

A priori N$_2$O emissions for anthropogenic, non-agricultural sources (including industrial processes, transportation, residential, and wastewater emissions) are from the Emission Database for Global Atmospheric Research (EDGARv4.2; http://edgar.jrc.ed.europa.eu), which are provided annually and total 2.3 Tg N yr$^{-1}$ for 2008. Monthly N$_2$O emissions from

non-agricultural soils are from CLMCN-N$_2$O as described by Saikawa et al. (2013). These emissions have been shown to accurately capture the magnitude and seasonality of soil emissions in the Amazon, but exhibited less skill in reproducing the observed seasonal cycle in northern midlatitudes (based on data from New Hampshire; Saikawa et al., 2013). The magnitude of these emissions varies depending on the meteorological forcing dataset used; forcings used here are from the MIT Integrated Global System Model (IGSM) fully coupled transient 20$^{th}$ century climate integration (Sokolov et al., 2009).

Together with the EDGARv4.2 agricultural emissions, this leads to an a priori global soil N$_2$O source of 11.0 Tg N yr$^{-1}$ for 2011. Biomass burning emissions are computed monthly based on the Global Fire Emissions Database version 3 (GFED3; van der Werf et al., 2010), totaling 0.6 Tg N yr$^{-1}$, while monthly oceanic N$_2$O emissions are from Jin and Gruber (2003) and total 3.5 Tg N yr$^{-1}$. The global annual a priori N$_2$O flux for 2011 is then 17.4 Tg N yr$^{-1}$, in the range of recent top-down estimates (16.1 to 18.7 Tg N yr$^{-1}$ for years 2006-2008; Saikawa et al., 2014; Thompson et al., 2014c). Stratospheric loss of

N$_2$O via photolysis and reaction with O($^1$D) is calculated from 3-D loss frequencies archived monthly from Global Modeling Initiative (GMI) simulations driven by MERRA meteorological fields; the resulting N$_2$O lifetime is ~127 years (note that the value depends on the initial spatial distribution of N$_2$O in the model).

The long N$_2$O lifetime necessitates accurate characterization of initial conditions to avoid biasing the optimized fluxes (e.g., Thompson et al., 2014c). In our work, we construct six sets of initial conditions from global N$_2$O observations and evaluate

the corresponding impacts on the inferred fluxes. Initial condition fields are constructed based on either data interpolation or 4D-Var optimization, with details discussed in Section 3.

### 2.2 Inversion frameworks

We employ three inversion methods with varying resolution to solve for N$_2$O emissions over two years (April 2010 – April 2012) based on global surface observations. The first of these is a 4-D Var inversion that iteratively optimizes emissions on

the native model grid (here 4° × 5°) using gradients computed with the GEOS-Chem adjoint model. This has the advantage of avoiding any aggregation errors associated with traditional clustering methods. However, our previous work (Wells et al., 2015) has shown that the degrees of freedom for atmospheric N$_2$O inversions is typically much less than the native model



grid dimension, and furthermore that native resolution optimizations have limited ability to resolve any temporal (e.g., seasonal) $N_2O$ emission biases. We therefore apply two alternate approaches to reduce the dimension of the inverse problem: 1) a 4D-Var inversion solving for emissions on aggregated, geographically-defined land and ocean regions, and 2) a 4D-Var inversion solving for emissions on a reduced emission basis set defined using an SVD-based information content analysis. In all three frameworks we consider two emission sectors (terrestrial and oceanic), and optimize monthly fluxes. We present details for each of the three frameworks in the following sections.

### 2.2.1 Standard 4D-Var inversion

Our standard inversion is a 4D-Var optimization in which the state vector contains scaling factors for monthly $N_2O$ emissions at $4° \times 5°$. The optimal set of emission scaling factors is obtained by minimizing the cost function, $J(p)$, which is a scalar containing contributions from the error-weighted model-measurement mismatch and the departure from the a priori values:

$$J(x) = \frac{1}{2}\sum_{c\in\Omega}(h(x) - y)^T S_y^{-1}(h(x) - y) + \frac{1}{2}(x - x_a)^T S_a^{-1}(x - x_a) , \qquad (1)$$

where $x$ is a vector of the parameters to be optimized (in this case, emission scaling factors), $x_a$ contains the a priori values of those parameters, $y$ is a set of observed $N_2O$ mixing ratios, $h(x)$ is a vector containing the simulated mixing ratios at the time and location of each observation, $S_y$ and $S_a$ are the observational and a priori error covariance matrices, and $\Omega$ represents the time-space domain of the observations.

We find the cost function minimum by using a quasi-Newton routine (Zhu et al.,1994; Byrd et al., 1995) to iteratively converge to $\min(J(x))$. At each iteration, we compute the gradient of $J(x)$ with respect to the emission scaling factor using the adjoint of GEOS-Chem. We use a lower bound of zero to avoid negative emission scaling factors and an upper bound of 10 based on our earlier work (Wells et al., 2015). The GEOS-Chem adjoint has previously been applied to a wide range of inverse problems for atmospheric composition, including constraining sources and sinks of long-lived greenhouse gases such as $CO_2$ (Deng et al., 2014; Liu et al., 2014; Deng et al., 2015, Liu et al., 2015), methane (Wecht et al., 2014; Turner et al., 2015), and $N_2O$ (Wells et al., 2015), as well as aerosols and reactive trace gases (e.g., Henze et al., 2007; Kopacz et al., 2009; Wells et al., 2014).

A priori uncertainties are assumed to be 100% for both land and ocean emissions, with off-diagonal terms assuming correlation length scales of 500 and 1000 km, respectively, following prior work by Thompson et al. (2011; 2014a). Observational errors are calculated as the quadratic sum of measurement uncertainty (~0.4 ppb for most sites, see Section 2.4) and model transport uncertainty, with the latter estimated from the 3-D model variance in $N_2O$ mixing ratios in the grid boxes surrounding any given observation (resulting in a mean uncertainty ~0.2 ppb at the surface). The solution presented here was calculated using 40 iterations, after which the cost function change per iteration is <1% and the total cost function reduction is ~65% (Fig. S2).



### 2.2.2 Continental-scale inversion

While the above approach avoids any aggregation error, the existing observational network provides insufficient information to constrain N$_2$O emissions in every 4° × 5° model grid square. Therefore, in an alternate inversion, we reduce the dimension of the inverse problem by solving for emission scaling factors on six continental (North America, South America, Europe,

Africa, Asia, Oceania) and three ocean regions (northern oceans: 30° – 90° N, tropical oceans: 30° S – 30° N, and southern oceans: 30° – 90° S). Regions are mapped in Fig. S1 and are similar to those used in the TransCom N$_2$O model intercomparison study (Thompson et al., 2014b; 2014c), except with one rather than two Asian regions. While this inversion could readily be carried out analytically rather than numerically (owing to its small dimension) we instead use 4D-Var for consistency and to impose the same scaling factor bounds (0-10) as in the standard inversion. We thus use the GEOS-Chem

adjoint to calculate the cost function gradient ($\partial J/\partial x$) aggregated over the 9 predefined regions. We then iteratively minimize $J(x)$, achieving a cost function change of < 1% per iteration (and total reduction of ~55%) after 28 iterations (Fig. S2).

### 2.2.3 SVD-based inversion

As an advance over standard aggregation methods such as the one described above, we also apply a new, efficient SVD-based information content analysis technique that maximizes the degrees of freedom of the inverse system while permitting

us to solve for N$_2$O fluxes in a fast iterative framework. The method, based on synthesis and advancement of recent works in this area (Flath et al., 2011; Bui-Thanh et al., 2012; Spantini et al., 2015) by Bousserez and Henze (2017), uses an optimal low-rank projection of the inverse problem that maximizes the observational constraints. Specifically, for a given dimension $k$, the optimal reduced space (Spantini et al., 2015; Bousserez and Henze, 2017) is spanned by the first $k$ eigenvectors of the prior-preconditioned Hessian $\mathbf{G}$ (Flath et al., 2011):

$$\mathbf{G} \equiv \mathbf{S}_a^{\frac{1}{2}} \mathbf{H}^T \mathbf{S}_y^{-1} \mathbf{H} \mathbf{S}_a^{\frac{1}{2}} = \mathbf{V} \mathbf{\Lambda} \mathbf{V}^T, \qquad\qquad\qquad (2)$$

where $\mathbf{H}$ is the tangent linear of the forward model, $\mathbf{V}$ is a matrix whose columns are the eigenvectors of $\mathbf{G}$, and $\mathbf{\Lambda}$ is a diagonal matrix containing the eigenvalues of $\mathbf{G}$. The following analytical approximation can then be used:

$$\mathbf{S}_{opt} = \mathbf{S}_a - \mathbf{S}_a^{\frac{1}{2}} \left( \sum_{i=1}^{k} \frac{\lambda_i}{\lambda_i+1} \boldsymbol{v}_i \boldsymbol{v}_i^T \right) \mathbf{S}_a^{\frac{1}{2}}, \qquad\qquad\qquad (3)$$

where $\mathbf{S}_{opt}$ is the posterior error covariance matrix, while $\boldsymbol{v}_{i,i=1,...,k}$ and $\lambda_{i,i=1,...,k}$ are the eigenvectors and eigenvalues of $\mathbf{G}$. This

expression gives, in some sense, the lowest error rank-$k$ approximation of $\mathbf{S}_{opt}$ (see Bousserez and Henze (2017) for details). The eigenvectors $\boldsymbol{v}_i$ can be interpreted as the most constrained modes in flux space, i.e. flux patterns that are independently constrained by the observations (Cui et al., 2014; Bousserez and Henze, 2017). These eigenvectors of the prior-preconditioned Hessian are efficiently calculated using a fully-parallelized randomized algorithm (Halko et al., 2011), as in Bui-Thanh et al. (2012) and Bousserez and Henze (2017). We use $k$=350 here, which employs nearly all modes with

eigenvalues greater than 1.0 (Fig. S3), as modes with eigenvalues below this threshold are informed mainly by the prior.



From $\mathbf{S}_{opt}$ we can obtain the inversion averaging kernel, which gives a measure of how well emissions are constrained in a given location, as follows:

$$\mathbf{AK} = \mathbf{I} - \mathbf{S}_{opt}\mathbf{S}_a, \tag{4}$$

where $\mathbf{I}$ is the identity matrix and $\mathbf{S}_a$ is the a priori error covariance matrix. Optimized solutions in areas where the diagonal

of $\mathbf{AK}$ is close to 1.0 are well-constrained by the observations. The trace of the averaging kernel gives the total degrees of freedom, i.e. the number of independent pieces of information that can be obtained in the inversion framework.

The posterior mean estimate of $x$ can also be directly calculated from analytical formulas using the eigenvectors of $\mathbf{G}$ (Spantini et al., 2015; Bouserez and Henze, 2017). However, to impose a positivity constraint on the emissions, we rely here on the variational minimization framework as in the standard 4D-Var case. In order to leverage the use of the optimal basis

set, we project both the cost function and its gradient onto the principal modes to obtain a reduced analytical formulation. The analytical expression for the reduced cost function (derivation presented in Appendix A) is:

$$J(x) \approx \frac{1}{2}(x - x_a)^T \mathbf{S}_a^{-\frac{1}{2}} \sum_{i=1}^{k} v_i v_i^T \mathbf{S}_a^{-\frac{1}{2}}(x - x_a) + \frac{1}{2}(h(x_a) - y)^T \mathbf{S}_y^{-1}(h(x_a) - y) + \frac{1}{2}(x - x_a)^T \mathbf{S}_a^{-\frac{1}{2}} \sum_{i=1}^{k} \lambda_i v_i v_i^T \mathbf{S}_a^{-\frac{1}{2}}(x - x_a) + \frac{1}{2}(h(x_a) - y)^T \mathbf{S}_y^{-\frac{1}{2}} \sum_{i=1}^{k} \lambda_i^{\frac{1}{2}} w_i v_i^T \mathbf{S}_a^{-\frac{1}{2}}(x - x_a) + \frac{1}{2}(x - x_a)^T \mathbf{S}_a^{-\frac{1}{2}} \sum_{i=1}^{k} \lambda_i^{\frac{1}{2}} v_i w_i^T \mathbf{S}_y^{-\frac{1}{2}}(h(x_a) - y), \tag{5}$$

while the analytical approximation for the cost function gradient is:

$$\nabla J(x) \approx \mathbf{S}_a^{-\frac{1}{2}} \sum_{i=1}^{k} v_i v_i^T \mathbf{S}_a^{-\frac{1}{2}}(x - x_a) + \mathbf{S}_a^{-\frac{1}{2}} \sum_{i=1}^{k} \lambda_i v_i v_i^T \mathbf{S}_a^{-\frac{1}{2}}(x - x_a) + \mathbf{S}_a^{-\frac{1}{2}} \sum_{i=1}^{k} \lambda_i^{\frac{1}{2}} v_i w_i^T \mathbf{S}_y^{-\frac{1}{2}}(h(x_a) - y), \tag{6}$$

where $k$=350 is the number of modes retained in the approximation. Here, $h(x_a)$ are the model mixing ratios corresponding to the a priori emissions and $w_i$ are the eigenvectors in observation space:

$$w_i = \frac{1}{\sqrt{\lambda_i}} \mathbf{S}_y^{-\frac{1}{2}} \mathbf{H} \mathbf{S}_a^{\frac{1}{2}} v_i. \tag{7}$$

Because the cost function and gradient depend only on the a priori model-measurement difference, the a priori and

observational error covariances, and the eigenvectors of $\mathbf{G}$ (which are computed only once), this iterative inversion offers significant time savings, particularly for models with a low level of parallelization. Monthly N$_2$O emission scaling factors for the 2-year analysis window are derived in approximately 6 hours, versus over 100 hours for the standard and continental-scale inversions, and nearly all the computation time in the former case is spent on calculating the eigenvectors of $\mathbf{G}$. The solution for the SVD-based inversion (with a projected cost function change of $< 1\%$ per iteration) is obtained after 60

iterations (Fig. S2). The full cost function reduction (calculated from a forward model run) is ~25% for this solution, whereas we achieve the minimum in the full cost function at a much earlier iteration (see Fig. S2). The divergence in the behavior of the projected and full cost function after this point may suggest that the weaker modes included here are not as well-approximated by the randomized SVD calculation as the dominant modes. An objective criteria for determining the error in the randomized SVD is the subject of a work in progress.



### 2.3 Atmospheric N$_2$O observations

Atmospheric N$_2$O observations used in our analysis include a global ensemble of surface measurements as well as airborne data from the HIAPER Pole-to-Pole Observations (HIPPO) campaigns (Wofsy, 2011). Because we found in our prior work that the surface dataset provides the strongest constraint on the spatial distribution of N$_2$O emissions (Wells et al., 2015), we

employ these in the inversion and reserve the airborne data for a posteriori evaluation.

Fig. 1 shows a map of the surface measurement sites used in this study. The surface measurements consist primarily of discrete air-filled flasks from NOAA's Cooperative Global Air Sampling Network (CCGG) program (Dlugokencky et al., 1994); we also include flask-based air samples from the Commonwealth Scientific and Industrial Research Organisation (CSIRO) network, the Environment Canada (EC) network, and a National Institute of Water and Atmospheric research

(NIWA) site. We assume a measurement uncertainty of 0.4 ppb at all flask sampling sites based on recommendations from the data providers. In addition to the flask-based air samples, we use high-frequency N$_2$O measurements (discrete hourly or hourly averaged) from the NOAA Chromatograph for Atmospheric Trace Species (CATS) network (Hall et al., 2007), the Advanced Global Atmospheric Gases Experiment (AGAGE) network (Prinn et al., 2000), and the University of Minnesota tall tower Trace Gas Observatory (TGO; Griffis et al., 2013; Chen et al., 2016). The hourly measurement uncertainty at these

sites is approximately 0.3 ppb, 0.6 ppb, and 1 ppb, respectively.

Small calibration offsets between measurement networks can significantly impact N$_2$O inversions due to its low ambient variability relative to background mixing ratios. To address this, we adjust here the AGAGE and EC data to the same NOAA 2006A scale used by the NOAA CCGG, CATS, CSIRO, NIWA, and TGO measurements. For AGAGE, we calculate an adjustment factor based on co-located CCGG flask-based air samples taken within 15 minutes of an in situ measurement at

five sites: CGO (Cape Grim, Australia), MHD (Mace Head, Ireland), RPB (Ragged Point, Barbados), SMO (Tutuila, American Samoa), and THD (Trinidad Head, California). The mean CCGG:AGAGE ratio at these sites from 2010 to 2012 is 1.00037, and we apply this adjustment to all AGAGE data. For EC, we calculate an adjustment factor based on co-located NOAA flask-based air measurements at ALT (Alert, Nunavut). The mean NOAA:EC ratio during our analysis period is 1.00017, and we use this adjustment factor across the EC network. While calibration scale offsets can be concentration- and

time-dependent, our relatively short (2-year) analysis window avoids the need for any temporally resolved measurement adjustments. Prior to our analysis we also screen for outliers by omitting any measurements more than two standard deviations (calculated on a running basis with a 30-day time window for flask-based air measurements and a 24-hour time window for in situ observations) away from its nearest neighbor.

For a posteriori evaluation of the inverse modeling results we employ airborne measurements from the HIPPO campaigns

(Wofsy, 2011), which featured pole-to-pole sampling and regular vertical profiling from approximately 300 to 8500 m altitude, with some profiles extending to 14000 m. Figure 1 shows flight tracks for the two deployments occurring during our simulation period and used here: HIPPO IV (June-July 2011) and HIPPO V (August-September 2011). The aircraft payload included high-frequency N$_2$O measurements by quantum cascade laser spectroscopy (QCLS; Kort et al., 2011). To ensure





calibration consistency we apply an offset adjustment to these data for each deployment based on concurrent flask-based air samples taken using an onboard Whole Air Sampler (WAS).

### 3 Inversion sensitivity to initial conditions for N$_2$O

Because of the ~127 year atmospheric lifetime for N$_2$O, any bias in the model initial conditions can persist throughout the
analysis period and lead to substantial errors in top-down emission estimates (Thompson et al., 2014c). In this section, we
evaluate six alternate approaches to generating initial N$_2$O mass fields for the start date of our inversions (1 April 2010), their
impact on the derived fluxes, and their overall suitability for inverse modeling.

The six treatments are summarized in Table 1. Three involve interpolation of surface observations from the NOAA,
AGAGE, CSIRO, EC, and NIWA networks for alternate time windows (MarZonal, AprZonal, AprKriging), and three
involve 4D-Var adjoint optimization of the initial mass field based on those same observations plus those from TGO
(AprOpt, FebOpt, RemoteOpt). Interpolation of observations offers the advantage of avoiding any model information that
may bias the initial state, whereas a 4D-Var optimization of the initial conditions allows us to exploit subsequent
atmospheric transport to inform the initial state in locations without N$_2$O observations. The first three approaches employ
either linear interpolation of zonally-averaged surface measurements or Kriging, and use observations from March 2010
(with subsequent one month model spin-up) or from 25 March to 7 April 2010 (with no subsequent spin-up). In each case,
the resulting surface mixing ratios of N$_2$O (mapped in Fig. S4) are assigned to all vertical levels in the troposphere; initial
N$_2$O mixing ratios above 100 hPa are based on interpolated mean profiles from the EOS Aura Microwave Limb Sounder
(MLS; Lambert et al., 2007). Where necessary, N$_2$O mixing ratios above the tropopause but below 100 hPa are linearly
interpolated between the tropospheric and MLS values.
The three tests in which the initial conditions are optimized by 4D-Var use a time window of February-March 2010, April-
May 2010, or January-June 2010 to solve for the initial N$_2$O mass field on 1 April 2010. Two of these assimilate all surface
observations while one employs only data from remote sites. Below, we evaluate each of the six initial condition treatments
against observations at the beginning of the simulation period (1-7 April 2010) and perform a standard 4D-Var optimization
of N$_2$O emissions to quantify the sensitivity of the inferred fluxes to the selected initial conditions.
Table 2 shows the initial bias statistics with respect to the surface observations for each initial condition treatment. Of the
interpolation approaches, the MarZonal setup has the poorest performance, with an overly strong interhemispheric gradient
and the largest initial model:measurement bias at all sites. In this case, the 1-month model spinup, meant to smooth out any
artificial N$_2$O gradients from the interpolation, is counterproductive as it allows model emission biases to accumulate prior to
the inversion. The interpolation methods without subsequent spinup (AprZonal, AprKriging) perform better in terms of
initial model:measurement bias and the interhemispheric N$_2$O gradient. We see the same general behavior when using 4D-
Var to optimize the initial conditions, with the no-spinup AprOpt approach providing the lowest initial model:measurement





bias (and least spread in bias) across all of the six methods tested. Using only data from remote sites (RemoteOpt) in the initial field optimization leads to a negative model bias, on average, in both hemispheres.

The bias statistics above can only test the realism of the initial $N_2O$ fields in those locations where there are observations, and say nothing about any potential bias in the large majority of model grid squares that lack observations. On the other

hand, by carrying out a full forward model run based on each of those initial conditions, we can exploit atmospheric transport to more fully assess the fidelity of the initial $N_2O$ mass field based on the evolution of model:measurement biases at the various observation sites.

Fig. 2 shows monthly-mean model-measurement residuals (averaged for Northern and Southern Hemisphere sites) for a full two-year forward simulation using the a priori emissions for each of the above initial mass fields. While most of the initial

conditions exhibit minimal bias at the start of the simulation, some develop large biases over time. As a result, the corresponding a posteriori global flux obtained in a 4D-Var source inversion (values shown inset in Fig. 2) varies considerably depending on the initial $N_2O$ field, with the flux adjustment even changing sign: a posteriori values range from 16.1 to 21.4 Tg N yr$^{-1}$, i.e. from a ~7% reduction to a 23% increase in the prior flux. We see in Fig. 2 that the direction of the global flux adjustment corresponds to the trend in the model-measurement residuals. For example, with the MarZonal initial

conditions, a significant negative trend in the residuals drives a global flux increase relative to the a priori, despite the fact that this case exhibits a positive mean bias with respect to the observations at the outset (Table 2). Such a trend in the model:measurement residuals could theoretically arise from incorrect initial conditions or from the accumulation of model source/sink errors over the course of the simulation. However, because our a priori flux and lifetime are broadly consistent with independent observational constraints (Prather et al., 2012), whereas an annual $N_2O$ source of 20+ Tg N would yield a

higher-than-observed atmospheric growth rate, a biased initial mass field is the more tenable explanation.

Overall, the three simulations using initial conditions optimized by 4D-Var yield a relatively small trend in the model-measurement $N_2O$ residuals, as does the AprZonal simulation, arguing for a more realistic initial $N_2O$ distribution in these cases. While the a posteriori flux between them varies, differences are less than 10% of the a priori flux. Because the AprOpt initial conditions exhibit the lowest initial bias, along with the lack of a trend in the residual timeline, we choose this method

to construct the initial conditions for the $N_2O$ inversions presented here. Likewise, for future work on $N_2O$ and other long-lived species, we recommend constructing the initial conditions by 4D-Var assimilation of observations at the outset of the inversion period. Because they are used for initial condition optimization, the April-May 2010 surface observations are excluded from the subsequent source inversions.

**4 Inversion evaluation and results**

Figure 3 shows maps of annual a posteriori $N_2O$ emissions from the standard, continental-scale, and SVD-based inversion for 2011, along with bar charts of the 2011 annual flux for the nine regions considered in the continental-scale inversion (numerical values listed in Table 3). A priori emissions are also included for comparison. We focus on 2011 results to





minimize any residual bias from the initial conditions. Focusing on 2011 also excludes the last three months of the inversion window, as the adjoint forcing weakens towards the end of the simulation due to the long lifetime of $N_2O$ (Wells et al., 2015).

The optimized global fluxes, listed inset in each map in Fig. 3, range from 15.9 Tg N yr$^{-1}$ for the SVD-based inversion to

17.5 – 17.7 Tg N yr$^{-1}$ for the standard and continental-scale inversions, with some similar spatial patterns and some discrepancies that we explore further in Section 4.3. The SVD-based global flux agrees well with that implied by its atmospheric lifetime and global burden for 2010 (15.7 ± 1.1 Tg N yr$^{-1}$; Prather et al., 2012). It also gives a better comparison to HIPPO measurements (see below), and thus appears to provide the best estimate of the true global flux. However, all three a posteriori global annual fluxes are within the range of recent inverse studies (16.1-18.7 Tg N yr$^{-1}$). Below we evaluate our

inversion results using aircraft and surface observations before interpreting them in terms of the information they provide on $N_2O$ emission processes.

**4.1 A posteriori evaluation of $N_2O$ emissions**

We apply the HIPPO IV and V airborne measurements described in Section 2.4 (and mapped in Fig. 1) to evaluate the a posteriori fluxes from our different inversion methods, and assess which method yields the most realistic depiction of true

$N_2O$ fluxes. Figure 4 shows average vertical profiles of the model-measurement $N_2O$ difference for these deployments in the a priori and the three inverse estimates (standard 4D-Var, continental-scale, and SVD-based inversions) as a function of latitude. Initially, the model vertical profile is biased high throughout the troposphere in the northern mid-to-high latitudes; this bias is larger during HIPPO V than HIPPO IV due to a seasonal bias in model emissions that is further discussed in Section 4.4. In the southern mid-to-high latitudes the model is also biased high through most of the troposphere. In most

cases in Fig. 4 we see that the model-measurement difference trends negative with height in the troposphere, which may reflect a model underestimate of the convective transport of $N_2O$ emissions (Kort et al., 2011). Large biases above 400 hPa in HIPPO IV (30° to 90° N) and HIPPO V (30° to 90° S) are driven by high latitude observations in which the aircraft is sampling below the model tropopause but above the actual tropopause, and highlight the difficulty in modeling the $N_2O$ vertical profile at these altitudes.

All three inversions significantly reduce the 30° to 90° N bias seen for both HIPPO campaigns; the SVD-based approach provides the fullest correction during HIPPO V, while slightly overcorrecting the HIPPO IV bias. However, the high bias from 30° to 90° S is only reduced in the SVD-based inversion, despite the fact that the continental-scale inversion has the lowest a posteriori emissions in this latitude range (Table 3). The lower global flux obtained with the SVD-based approach (Fig. 3 and Table 3) is thus the reason for this correction, implying that the global annual a priori flux is too high. We note

that a slight low bias does emerge in the tropics in the SVD-based approach, where the spatial distribution of emissions is particularly difficult to resolve.




### 4.2 Averaging kernel

The information from the randomized-SVD algorithm can be used to directly calculate the inversion averaging kernel (AK) and posterior error via Eqs. (3) and (4), giving valuable information on the spatial distribution of emission constraints provided by the $N_2O$ observing network. Figure 5 shows the diagonal of the AK for $N_2O$ emissions in April 2011 (results for

other months are very similar). AK diagonal values near 1.0 indicate emission locations that are well-constrained by observations, while AK diagonal values close to 0 indicate emission locations that lack a direct constraint.

AK diagonal values for monthly $N_2O$ emissions are highest in the US and Europe where the observational coverage is most extensive, with values up to 0.7 in locations where hourly observations are available. Weaker constraints are achieved in East Asia and some tropical and Australian grid boxes, with AK values ranging from 0.01-0.4. AK values throughout most

of the Tropics, Southern Hemisphere, Canada, and northern Asia reveal almost no direct observational constraints on monthly emissions in these regions.

The number of pieces of information that can be independently resolved (DOFs) in any inversion can be determined from the trace of the AK. Here, the DOFs are ~315 for the full two-year inversion. A key advantage of the SVD-based approach is that it solves for only those spatiotemporal flux patterns that can be constrained by the observations: i.e., the dimension of

the solution is consistent with the DOFs of the inversion. On the other hand, the standard inversion attempts to resolve 79,466 free variables, ~250× more than can legitimately be constrained, while the continental-scale inversion yields fewer pieces of information (216) than are obtainable. The latter point confirms that the observations can in fact resolve some finer-scale spatial and temporal information on $N_2O$ emissions in the regions where AK values are highest.

### 4.3 Regional annual $N_2O$ emissions

In this section we interpret the inversion results by region in terms of their implications for present understanding of $N_2O$ emission processes. We focus on the spatial information obtained from the standard and SVD-based inversions and on those features that are most robust across these inversion frameworks.

#### 4.3.1 North America

A posteriori emissions from North America range from 1.24-1.78 Tg N yr$^{-1}$, with a slight increase (11%) inferred relative to

the a priori inventory for the continental-scale inversion versus a 20-23% decrease for the standard and SVD-based inversion. The latter values are quite close to a recent estimate from Saikawa et al. (2014) for 2008 (1.2 ± 0.2 Tg N yr$^{-1}$). Both the standard and SVD-based inversions call for a large increase (2-3×) in emissions from the US corn belt, one of the most intensively managed agricultural regions of the world. The magnitude of this upward adjustment supports emission underestimates previously found for this region (Kort et al., 2008; Miller et al., 2012; Griffis et al., 2013), and is likely due to

underrepresentation of the indirect $N_2O$ source associated with leaching and runoff from agricultural soils (Chen et al.,




2016). Emissions decrease in the western US and Canada (in both the standard and SVD inversions), where natural soil emissions may be too high in the CLMCN-N$_2$O inventory (Saikawa et al., 2014) used here as a priori.

### 4.3.2 South America

A posteriori emissions from South America range from 3.28-3.68 Tg N yr$^{-1}$, increasing 6-19% over the a priori. These values

are 40-60% larger than the median inferred by Thompson et al. (2014c) for 2006-2008 (2.33 Tg N yr$^{-1}$); however, due to weak observational constraints (Fig. 5) we find that the results here are quite sensitive to the inversion framework used. For example, including fewer modes in the SVD-based solution yields an even higher a posteriori flux in this region, and the spatial distribution of emissions differs substantially between the standard and SVD-based solutions. Saikawa et al. (2014) do note a large recent increase in nitrogen fertilizer consumption over this region (49% from 1995-2008), which may help

explain the larger a posteriori flux seen here, although N fertilizer use in this region was only 7% of the global total in 2011 (International Fertilizer Association, 2016).

### 4.3.3 Europe

All three inversions point to a significant model overestimate of European N$_2$O emissions, with a posteriori fluxes that are 38% (standard inversion; optimized flux 1.05 Tg N yr$^{-1}$) to 75% (SVD-based inversion; optimized flux 0.43 Tg N yr$^{-1}$) lower

than the a priori. These optimized fluxes are in better agreement with the other top-down flux estimates for Europe (both for 2006) of 1.19 Tg N yr$^{-1}$ (Corazza et al., 2011) and 0.93 ±0.12 Tg N yr$^{-1}$ (Saikawa et al., 2014). The European source derived in the SVD-based and continental-scale inversions (0.43-0.57 Tg N yr$^{-1}$) represents ~3% of the global flux found in each case, which agrees with the result from Huang et al. (2008). We find the largest emission reductions over western and central Europe, suggesting an overestimate of soil and non-agricultural anthropogenic sources in the EDGARv4.2 inventory used

here. While non-agricultural anthropogenic sources provide a small contribution to the global N$_2$O flux (~13%), they comprise ~40% of the a priori European emissions in the model. Furthermore, based on the spatial distribution of the adjustments derived in the standard and SVD-based adjustments, this source undergoes a larger relative reduction than the soil source when integrated over Europe as a whole.

### 4.3.4 Africa

Annual emissions from Africa range from 2.85-2.97 Tg N yr$^{-1}$ in all three inversions, an 8-12% increase from the prior flux. Our a posteriori values are closer to the median optimized African flux found by Thompson et al. (2014c) for 2006-2008 (3.36 Tg N yr$^{-1}$) than is the a priori; however, the lack of direct observational constraints for this region (Fig. 5) prevent any definitive conclusion. As in South America, the SVD-based result here is quite sensitive to the number of modes used, with emission increments differing in sign for some months. The spatial distribution between the standard and SVD-based

solutions also differs, with the former preserving the a priori distribution and the latter placing more of the flux in equatorial Africa.





### 4.3.5 Asia

Over Asia the a posteriori flux ranges from 3.82 Tg N yr$^{-1}$ (9% decrease from the a priori) to 4.59 Tg N yr$^{-1}$ (10% increase). The full-dimensional and SVD-based inversions both call for a reduction in model emissions for northern China and Russia and an increase to the south. Consistent a posteriori spatial patterns emerge in the latter region, with large emission increases

over the prior for the Indo-Gangetic Plain (IGP) of India, Southeast Asia, and Eastern China. Our flux estimates are towards the higher end of the wide range of estimates for North + South Asia (2.87-4.48 Tg N yr$^{-1}$) reported by Thompson et al. (2014c) for 2006-2008; that study concludes that observational constraints are low in this region, which is generally consistent with our findings (Fig. 5). Saikawa et al. (2014) find that agricultural N$_2$O emissions are increasing in South Asia, and that is consistent with our higher flux for 2011 compared to the Thompson et al. (2014c) median value for 2006-2008.

58% of global N fertilizer consumption occurred in South and East Asia in 2011 (International Fertilizer Association, 2016); it is possible that direct, on-field N$_2$O emissions here are underestimated due to the nonlinear increase in emissions as N inputs exceed crop demands (Shcherbak et al, 2014). Over northern Asia our results point to an overestimate of natural soil emissions (as this is the dominant regional source in the model); a similar overestimate was inferred by Saikawa et al. (2014) using the same a priori inventory.

### 4.3.6 Oceania

The emission estimates for Oceania range from 0.64 Tg N yr$^{-1}$ (16% decrease from the prior) to 0.84 Tg N yr$^{-1}$ (10% increase). Observational constraints are low in this region (outside of Cape Grim, where a measurement site exists, Fig. 5) and results depend strongly on the a priori. The emission reduction seen in the continental-scale inversion (Fig. 3) could also reflect a model overestimate of the southern ocean source, as the sparse observations make it difficult to separate land versus

ocean emissions here.

### 4.3.7 Ocean emissions

We obtain an annual flux ranging from 0.07-0.52 Tg N yr$^{-1}$ for northern oceans (30° – 90° N), 2.19-2.99 Tg N yr$^{-1}$ for tropical oceans (30° S – 30° N), and 0.39-0.70 Tg N yr$^{-1}$ for southern oceans (30° – 90° S). In all cases, our results indicate an emission increase for tropical oceans emissions (of 9-47%) and a decrease for northern (20-90%) and southern (11-51%)

oceans relative to the a priori Jin and Gruber (2003) inventory. The wide range of values reflects the degree to which the surface observing network can constrain ocean emissions. However, the standard and SVD-based inversions both point to a model overestimate in the North Atlantic where downwind observations in Europe have some power to resolve monthly emissions.

The direction of the oceanic emission changes is consistent with the findings of Thompson et al. (2014c); however, our

oceanic fluxes are lower than obtained in that study (1.08, 3.66, and 1.20 Tg N yr$^{-1}$ for northern, tropical, and southern oceans, respectively). Results obtained here (3.38-3.45 Tg N yr$^{-1}$) are more consistent with the most recent best estimate of





the oceanic source derived from global observations of the air-sea N$_2$O gradient (2.4 ± 0.8 Tg N yr$^{-1}$; Buitenhuis et al., 2017), albeit still higher. We find that ocean emissions make up ~20% of the global N$_2$O flux (in both the a priori and a posteriori estimates), lower than found in some inverse studies (31-38%; Saikawa et al., 2014; Thompson et al., 2014c) but consistent with Huang et al. (2008) (~23%).

### 4.3.7 Summary of regional scale results

Among the most robust spatial features of our results across all the inversion frameworks employed is an increase in annual N$_2$O emissions over the a priori in the tropics (particularly 0°-30° N), and a decrease at higher latitudes for both ocean and terrestrial sources. While the total Asian flux differs between the full-dimensional and SVD-based inversion, both solutions indicate a model overestimate in northern Asia and an underestimate in Southeast Asia. Furthermore, while the inversions disagree on whether the a priori emissions are too high or too low over North America as a whole, both the full-dimensional and SVD-based inversions increase the prior N$_2$O emissions over the US corn belt and reduce them over the western US and Canada. This suggests that while the a priori emissions may be too high in northern mid-to-high latitudes overall (which we attribute to overly-high natural soil emissions in the model, as well as non-agricultural anthropogenic emissions in regions such as Europe) they are underestimated for fertilized agricultural soils.

### 4.4 Seasonality of N$_2$O emissions

#### 4.4.1 A priori seasonality

Figure 2 shows that the a priori model bias in atmospheric N$_2$O varies strongly as a function of season in the Northern Hemisphere, implying a corresponding seasonal bias in the bottom-up emissions driving the model. Seasonality in our prior emissions is dominated by the natural soil source. Here, we compare the temporal constraints afforded by the different inversions, focusing again on the most robust features across the inversions, after first examining the seasonality differences between modeled and measured N$_2$O mixing ratios.

Figure 6 shows two-year timelines of monthly-averaged a priori modeled and measured N$_2$O mixing ratios along with the corresponding model-measurement residual for all surface measurement sites. The modeled N$_2$O from 30° – 90° N is characterized by a November – December peak, and a May – June minimum. This is out of phase with the measurements, which have a minimum around August – September and a peak in February – March. Several other CTMs in an a recent intercomparison (Thompson et al., 2014b; 2014c) likewise produce a seasonal minimum that is too early compared to observations, which that study suggests may reflect an overestimate of the impact of N$_2$O-depleted stratospheric air on surface mixing ratios. Our previous work indicates that surface N$_2$O mixing ratios are not sensitive to biases in the magnitude of the stratospheric sink on the timescale of our inversion (Wells et al., 2015), while Thompson et al. (2011) find that errors in modeled stratosphere-troposphere exchange (STE) can bias inferred regional emissions by up to 25%,



particularly over the North Atlantic and Europe. We thus focus here on inferred seasonal changes that are significantly larger than 25% and most robust to any potential errors in modeled STE.

Measured mixing ratios at the KCMP tall tower site in Minnesota are significantly higher than other Northern Hemisphere sites. As a result, it is one of the few sites where negative model-measurement residuals persist through most of the two-year
inversion period. The fact that it is also one of the only sites located in an agricultural source region provides support for previous findings of a missing or strongly underestimated agricultural $N_2O$ source tied to indirect emissions (Griffis et al., 2013; Chen et al., 2016).

### 4.4.1 Seasonality of $N_2O$ inversion results

Figure 7 contains 2011 timelines of the monthly a priori and a posteriori emissions for the three inversion methods over the
same continental and ocean regions considered above. Both North American and European a posteriori emissions are characterized by a shift from a summertime (June-July) to springtime peak in emissions (March-April), with the North American results exhibiting separate spring and summer peaks (plus a fall-winter enhancement in the SVD-based inversion). The a posteriori seasonality over Asia is nearly reversed from the a priori, with dual emission peaks in spring (March-April) and fall (September-October). This double maximum is consistent with past work and coincides with the approximate start
and end times of the Asian monsoon (Thompson et al., 2014c). Over South America and Africa we find that the a posteriori seasonality depends more strongly on the inversion method used, reflecting the low observational constraints in these regions (Fig. 5). Tropical ocean emissions increase primarily during summer and fall when emissions are at their peak, though the magnitude depends on the inversion method used. Emissions decrease strongly for the northern oceans (though they were not large to begin with) for the continental and SVD-based inversions, but with no shift in seasonality. Seasonal emission
adjustments are small over the southern oceans and Oceania, where constraints are weak.

The shift toward earlier springtime emissions in the Northern Hemisphere is one robust feature across our inversions. Thompson et al. (2014c) arrived at the same finding, and argued that it reflects the dependence of $N_2O$ emissions on soil moisture and temperature, as drier soils later in summer may limit $N_2O$ fluxes. However, other factors are also likely to contribute. Emissions associated with freeze-thaw cycles can lead to elevated springtime $N_2O$ fluxes at these mid- to high-
latitudes (e.g., Wagner-Riddle et al., 2017), while higher springtime emissions are also consistent with the timing of fertilizer application and indirect $N_2O$ emissions due to leaching and runoff from agricultural soils (Chen et al., 2016). The separate spring and summer emission peaks seen over North America in 2011 may reflect the respective influences of indirect and direct emissions, which have been shown (Chen et al., 2016) to peak earlier (indirect emissions) and later (direct emissions) in the growing season.

We see in Fig. 7 that the seasonal adjustments are larger in the continental and SVD-based inversion than the standard 4D-Var, particularly in regions where direct observational constraints are low. In our previous work (Wells et al., 2015) we highlighted the difficulty in correcting seasonal biases when solving for monthly $N_2O$ emissions on a grid box scale. The





SVD-based approach thus provides a major advantage in this context, by reducing the dimensions of the inverse problem and allowing us to better resolve temporal features that inform our understanding of $N_2O$ emission processes.

**5 Conclusions and implications for the $N_2O$ budget**

In this paper we employed three inversion frameworks to derive top-down constraints on global, monthly $N_2O$ emissions for
2011. The inverse frameworks included: (1) a standard 4D-Var inversion at 4° × 5°, (2) a 4D-Var inversion solving for fluxes on six continental and three ocean regions, and (3) a fast 4D-Var inversion based on a new dimension reduction technique using efficient randomized SVD algorithms. The latter technique is an advance over arbitrary aggregation schemes: the resolution of the solution is defined quantitatively according to the information afforded by the observations; the DOFs of the inverse system are maximized; and major time savings are achieved compared to other iterative inversion
methods.

- Over many regions, our inversion results are broadly consistent with other recent inversion studies, though the range of derived flux values and seasonalities from poorly-observed regions highlights the ill-posed nature of the inverse problem for $N_2O$. Based on the most robust features across our three different inversion frameworks, we can draw the following conclusions about the global $N_2O$ budget and underlying emission processes:

- The global annual $N_2O$ flux is likely somewhat high in the bottom-up inventory used here, as the lower value (15.9 Tg N yr$^{-1}$) derived in the SVD-based inversion gives a better representation of the $N_2O$ background while also being more consistent with the current best estimate from a 0-D consideration of the global burden and lifetime of $N_2O$ (Prather et al., 2012).

- Our inversion results indicate that a greater fraction of the global $N_2O$ flux is emitted from the tropics than the a
priori inventories would suggest. This points to an overestimate of natural soil (and perhaps industrial/residential) emissions in the Northern Hemisphere, and to an underestimate of agricultural (and likely oceanic) emissions in the tropics.

- In the Northern Hemisphere midlatitudes, $N_2O$ emissions peak earlier in the year (March-April) than our current inventories suggest (June-July). This seasonality is consistent with higher soil moisture and thawing of frozen soils
in the springtime, and with the timing of fertilizer application.

- We find that $N_2O$ emissions from agricultural soils are underestimated, and that this is likely due to an underestimate of indirect agricultural emissions during early spring and summer in the Northern Hemisphere. For example, annual emissions over the US Corn Belt are underestimated by 2-3× in the a priori inventories; the standard and SVD-based inversions used here both increase emissions from this region throughout the growing
period (March – September).

Based on our analysis of alternate initial conditions for atmospheric $N_2O$, and their corresponding effects on derived fluxes, we recommend formally optimizing the initial mass field (either alone or in tandem with the emissions optimization) rather





than interpolating N$_2$O observations or using an unconstrained model spinup. The impacts can be substantial: for the sensitivity tests used here, a posteriori global fluxes ranged by ~25% (16.1 – 21.4 Tg N yr$^{-1}$) across different treatments of the initial N$_2$O mass.

Finally, the SVD-based inverse approach used here offers a powerful framework for maximizing the emission information

derived from atmospheric observations of N$_2$O in an efficient, timely manner, particularly for models with a low level of parallelization. The approach provides valuable spatially-resolved information that is lost when solving for fluxes over ad-hoc continental-scale regions, while also providing a much stronger ability to resolve broad temporal features than is possible with a standard 4D-Var inversion at the model grid resolution. Such information is key to furthering our understanding of N$_2$O emission processes based on top-down analyses.

**Code availability**

The N$_2$O version of the GEOS-Chem adjoint code is available via the GEOS-Chem adjoint repository. Instructions for obtaining access to the code can be found at http://wiki.seas.harvard.edu/geos-chem/index.php/GEOS-Chem_Adjoint.

**Appendix A: Proof for cost function projection formula**

$$J(x) = \frac{1}{2}(h(x) - y)^T S_y^{-1}(h(x) - y) + \frac{1}{2}(x - x_a)^T S_a^{-1}(x - x_a),$$  (A1)

and

$$h(x) = h(x_a) + H(x - x_a),$$  (A2)

Therefore,

$$(h(x) - y)^T S_y^{-1}(h(x) - y) = (h(x_a) + H(x - x_a) - y)^T S_y^{-1}(h(x_a) + H(x - x_a) - y) = \left(h(x_a) + S_y^{\frac{1}{2}} S_y^{-\frac{1}{2}} H S_a^{\frac{1}{2}} S_a^{-\frac{1}{2}}(x - \right.$$

$$\left. x_a) - y\right)^T S_y^{-1}\left(h(x_a) + S_y^{\frac{1}{2}} S_y^{-\frac{1}{2}} H S_a^{\frac{1}{2}} S_a^{-\frac{1}{2}}(x - x_a) - y\right),$$  (A3)

Then we develop $S_y^{-\frac{1}{2}} H S_a^{\frac{1}{2}} = \sum_{i=1}^{n} \lambda_i w_i v_i^T$, where $n$ is the dimension of the state vector, and project the control variable onto the optimal basis $\{S_a^{\frac{1}{2}} v_i, i = 1, \dots, k\}$ using the projector $\pi = S_a^{\frac{1}{2}} \sum_{i=1}^{k} v_i v_i^T S_a^{-\frac{1}{2}}$, which yields:

$$(h(x) - y)^T S_y^{-1}(h(x) - y) \approx (h(x_a) - y)^T S_y^{-1}(h(x_a) - y) + (x - x_a)^T S_a^{-\frac{1}{2}} \sum_{i=1}^{k} \lambda_i v_i v_i^T S_a^{-\frac{1}{2}}(x - x_a) + (h(x_a) - $$

$$y)^T S_y^{-\frac{1}{2}} \sum_{i=1}^{k} \lambda_i w_i v_i^T S_a^{-\frac{1}{2}}(x - x_a) + (x - x_a)^T S_a^{-\frac{1}{2}} \sum_{i=1}^{k} \lambda_i v_i w_i^T S_y^{-\frac{1}{2}}(h(x_a) - y),$$  (A4)

and

$$(x - x_a)^T S_a^{-1}(x - x_a) \approx (x - x_a)^T S_a^{-\frac{1}{2}} \sum_{i=1}^{k} v_i v_i^T S_a^{-\frac{1}{2}}(x - x_a),$$  (A5)



Inserting Eqs. (A4) and (A5) in Eq. (A1), one obtains:

$$J(x) \approx \frac{1}{2}(x - x_a)^T S_a^{-\frac{1}{2}} \sum_{i=1}^{k} v_i v_i^T S_a^{-\frac{1}{2}}(x - x_a) + \frac{1}{2}(h(x_a) - y)^T S_y^{-1}(h(x_a) - y) + \frac{1}{2}(x - x_a)^T S_a^{-\frac{1}{2}} \sum_{i=1}^{k} \lambda_i v_i v_i^T S_a^{-\frac{1}{2}}(x -$$

$$x_a) + \frac{1}{2}(h(x_a) - y)^T S_y^{-\frac{1}{2}} \sum_{i=1}^{k} \lambda_i^{\frac{1}{2}} w_i v_i^T S_a^{-\frac{1}{2}}(x - x_a) + \frac{1}{2}(x - x_a)^T S_a^{-\frac{1}{2}} \sum_{i=1}^{k} \lambda_i^{\frac{1}{2}} v_i w_i^T S_y^{-\frac{1}{2}}(h(x_a) - y), \qquad (A6)$$

Differentiating Eq. (A6), one obtains:

$$\nabla J(x) \approx S_a^{-\frac{1}{2}} \sum_{i=1}^{k} v_i v_i^T S_a^{-\frac{1}{2}}(x - x_a) + S_a^{-\frac{1}{2}} \sum_{i=1}^{k} \lambda_i v_i v_i^T S_a^{-\frac{1}{2}}(x - x_a) + S_a^{-\frac{1}{2}} \sum_{i=1}^{k} \lambda_i^{\frac{1}{2}} v_i w_i^T S_y^{-\frac{1}{2}}(h(x_a) - y), \qquad (A7)$$

**Acknowledgements**

This work was supported by NOAA (grant no. NA13OAR4310086) and the Minnesota Supercomputing Institute. We thank
E. Kort and S. Wofsy for providing the HIPPO $N_2O$ measurements. We thank Environment Canada for providing data from
the Alert, Churchill, Estevan Point, East Trout Lake, Fraserdale, and Sable Island Sites. We thank R. Martin and S. Nichol
for providing data from the Arrival Heights NIWA station. We thank J. Muhle and C. Harth (UCSD-SIO), P. Fraser
(CSIRO), R. Wang (GaTech), and other members of the AGAGE team for providing AGAGE data. The AGAGE Mace
Head, Trinidad Head, Cape Matatula, Ragged Point, and Cape Grim stations are supported by NASA grants to the
Massachusetts Institute of Technology and Scripps Institution of Oceanography; the Department of Energy and Climate
Change (DECC, UK) contract to the University of Bristol; and by CSIRO and the Australian Bureau of Meteorology. We
thank C. Adam Schlosser for work on the MIT IGSM.

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



| Test Name | Observational time range | Sites | Estimation method | Spin-up |
|-----------|--------------------------|-------|-------------------|---------|
| MarZonal | 1 – 31 March 2010 | All | Zonal average, linear interp | One month |
| AprZonal | 25 March – 7 April 2010 | All | Zonal average, linear interp | None |
| AprKriging | 25 March – 7 April 2010 | All | Kriging | None |
| AprOpt | 1 April – 31 May 2010 | All | 4D-Var | None |
| FebOpt | 1 February – 31 March 2010 | All | 4D-Var | Two months |
| RemoteOpt | 1 January – 30 June 2010 | Remote | 4D-Var | Three months |

**Table 1: The six initial conditions (for 1 April 2010) tested for $N_2O$, including the time range of observations used, observation sites included, interpolation or optimization method used, and length of spin-up.**




| Test Name | Bias: All Sites (ppb) | | | Bias: NH Sites (ppb) | | | Bias: SH Sites (ppb) | | |
|---|---|---|---|---|---|---|---|---|---|
| | 25th | Median | 75th | 25th | Median | 75th | 25th | Median | 75th |
| MarZonal | -0.21 | 0.30 | 0.71 | 0.11 | 0.46 | 0.86 | -0.66 | -0.36 | -0.15 |
| AprZonal | -0.13 | 0.20 | 0.62 | -0.03 | 0.32 | 0.73 | -0.38 | -0.12 | 0.10 |
| AprKriging | -0.29 | 0.06 | 0.42 | -0.31 | 0.02 | 0.39 | -0.20 | 0.14 | 0.49 |
| AprOpt | -0.21 | 0.01 | 0.21 | -0.22 | 0.01 | 0.20 | -0.21 | 0.01 | 0.22 |
| FebOpt | -0.29 | 0.06 | 0.48 | -0.42 | -0.03 | 0.37 | -0.16 | 0.14 | 0.39 |
| RemoteOpt | -0.48 | -0.14 | 0.22 | -0.44 | -0.09 | 0.33 | -0.58 | -0.30 | -0.04 |

5    **Table 2: Initial bias statistics for each of the six initial conditions with respect to observations at all sites, NH sites, and SH sites. Statistics are calculated for the first week of the simulation (1-7 April 2010).**





| Region | A priori emissions | A posteriori emissions | | |
|---|---|---|---|---|
| | | Standard    4D-Var inversion | Continental-scale inversion | SVD-based inversion |
| North America | 1.61 | 1.30 | 1.78 | 1.24 |
| South America | 3.09 | 3.68 | 3.58 | 3.28 |
| Europe | 1.70 | 1.05 | 0.57 | 0.43 |
| Africa | 2.65 | 2.97 | 2.92 | 2.85 |
| Asia | 4.18 | 4.47 | 4.59 | 3.81 |
| Oceania | 0.76 | 0.79 | 0.64 | 0.84 |
| Northern oceans (30° - 90° N) | 0.66 | 0.52 | 0.07 | 0.15 |
| Tropical oceans (30° S - 30° N) | 2.03 | 2.19 | 2.99 | 2.70 |
| Southern oceans (30° - 90° S) | 0.79 | 0.70 | 0.39 | 0.53 |
| **Global** | **17.4** | **17.7** | **17.5** | **15.9** |

5 **Table 3: 2011 N$_2$O emissions (Tg N yr$^{-1}$) over six continental and three oceanic regions for the a priori database, and a posteriori results for the three inversion frameworks used here.**





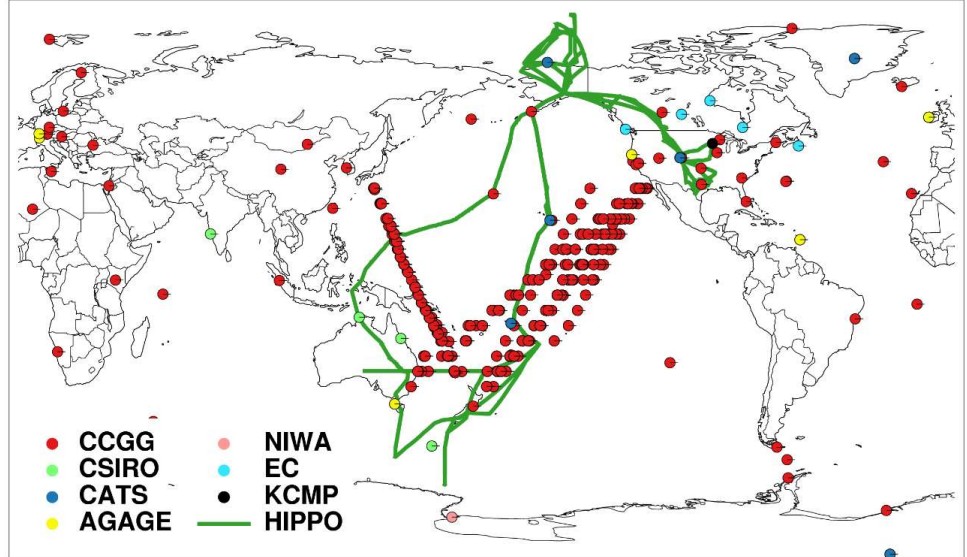

**Figure 1: Global surface observing network for atmospheric N₂O. Shown are surface discrete measurement locations for the NOAA Carbon Cycle and Greenhouse Gases (CCGG) network, the Commonwealth Scientific and Industrial Research Organisation (CSIRO) network, the National Institute of Water and Atmospheric Research (NIWA) network, and the Environment Canada (EC) network, as well as semi-continuous measurement locations in the NOAA Chromatograph for Atmospheric Trace Species (CATS) network, the Advanced Global Atmospheric Gases Experiment (AGAGE) network, and the KCMP tall tower site. Also shown are flight tracks from the HIPPO IV and V deployments.**



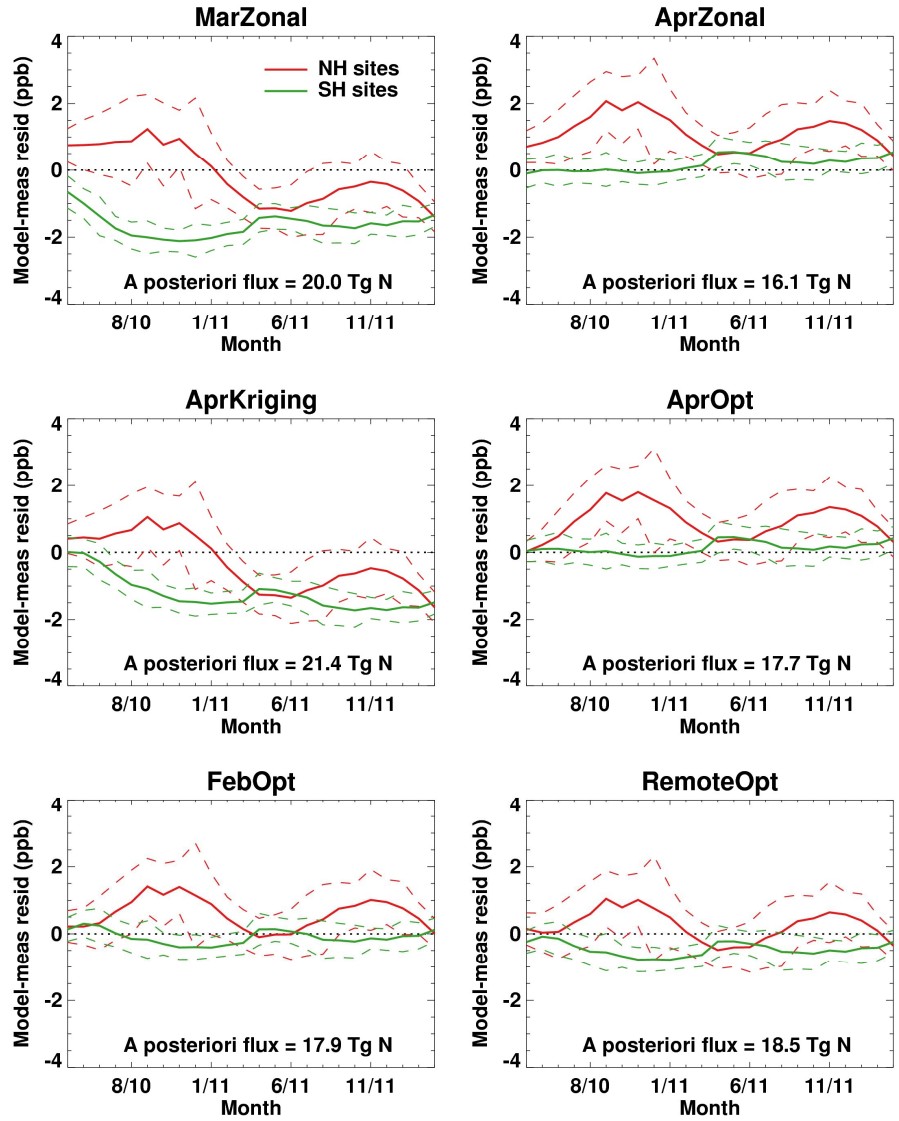

**Figure 2: Impact of initial conditions on a two-year N$_2$O simulation and inversion. Shown are timelines of the model-measurement residuals for a two-year forward-model simulation initialized using each of the six initial conditions listed in Table 1. The solid line represents the mean and the dashed lines represent the standard deviation about the mean for Northern Hemisphere (red) and Southern Hemisphere sites (green). The final 2011 a posteriori global flux for each simulation derived using a standard 4D-Var inversion is noted at the bottom of each panel.**





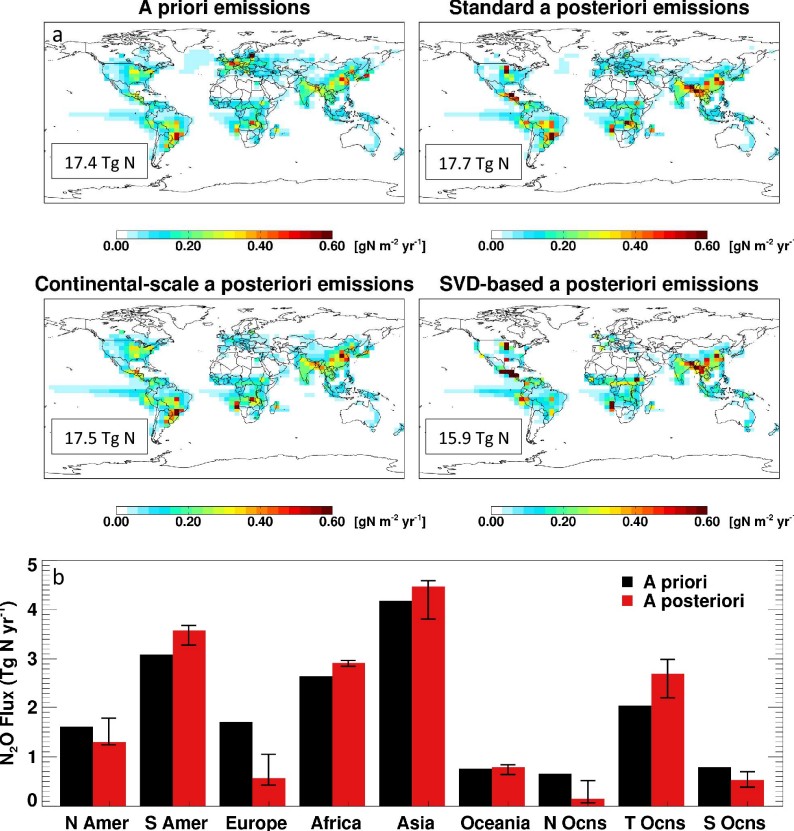

**Figure 3: (a) 2011 annual N₂O emissions (gN m⁻² yr⁻¹) for the a priori database and a posteriori results for each of the inversion frameworks used here (standard 4D-Var, continental-scale inversion, SVD-based inversion). Global fluxes are shown inset in each map. (b) 2011 annual N₂O flux (Tg N yr⁻¹) over six continental and three oceanic regions for the a priori database (black), and the a posteriori median from the three inversion frameworks (red). Error bars denote the range of a posteriori values for each region.**



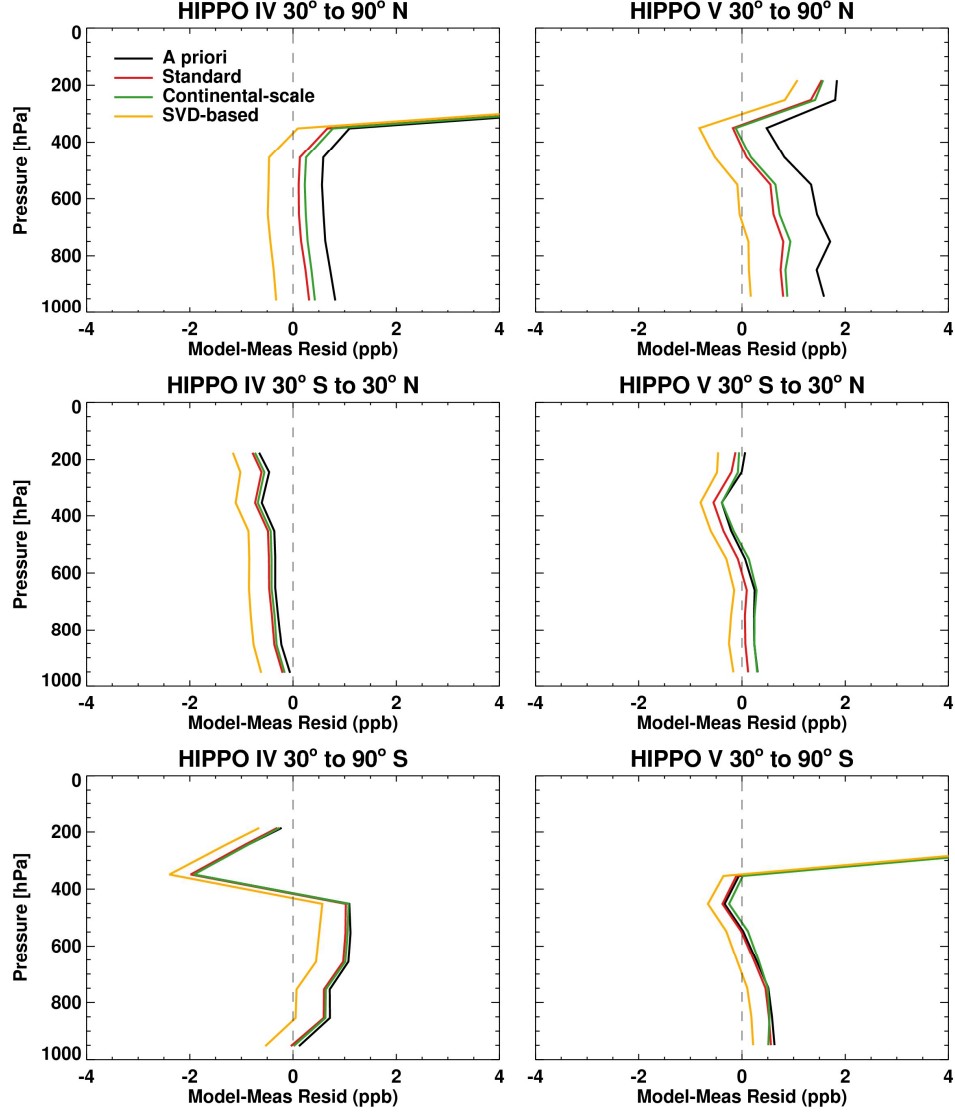

**Figure 4: A posteriori evaluation of N₂O inversion results using HIPPO data (not themselves used in the inversion). Shown are average vertical profiles of the model-measurement difference for HIPPO IV (14 June-11 July 2011, left column) and HIPPO V (9 August-9 September 2011, right column) as a function of latitude. A priori results are shown in black and a posteriori results in red (standard 4D-Var inversion), green (continental inversion), and gold (SVD-based inversion).**





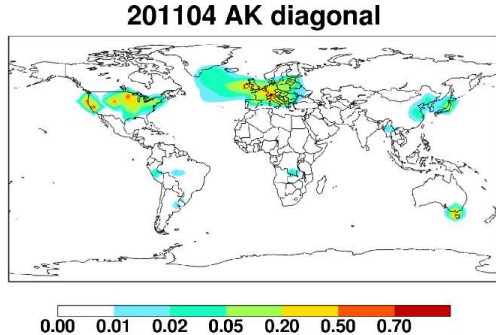

**Figure 5: Averaging kernel diagonal values for April 2011 in the SVD-based inversion, calculated from Eq. (4).**





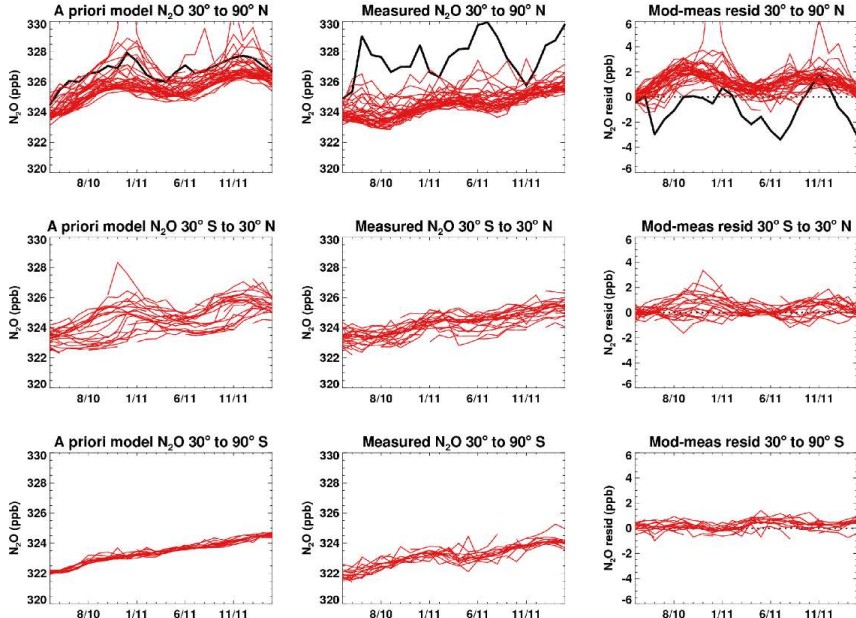

**Figure 6: Two-year timelines of monthly-averaged a priori modeled and measured N₂O mixing ratios, and the resulting model-measurement residuals, for individual measurement sites as a function of latitude. The solid black line in the top panels shows results for the KCMP tall tower site in Minnesota.**





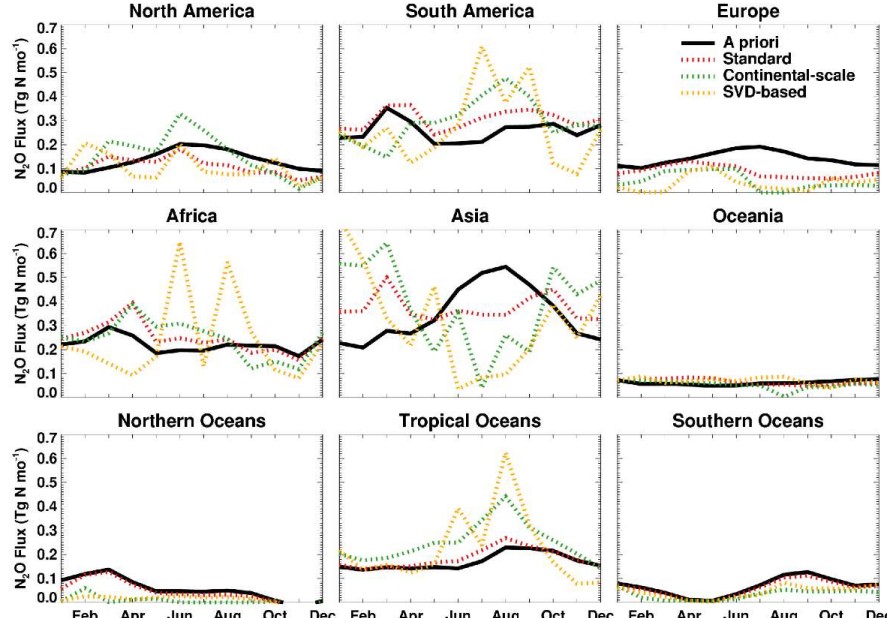

**Figure 7: Monthly N₂O emissions (Tg N month⁻¹) for 2011 over six continental and three oceanic regions. Shown is the a priori database (black), and a posteriori results for the standard 4D-Var inversion (red), the continental-scale inversion (green), and the SVD-based inversion (gold).**

