# Peer review of "Top-down constraints on global N2O emissions at optimal resolution: application of a new dimension reduction technique"

_Atmospheric Chemistry and Physics, 2017_

## Referee Comment (RC1) · Anonymous Referee #1 · 11 Sep 2017

The manuscript by Welles et al address an important need which is the comparison of top-down and bottom-up estimates of global N2O emissions. The authors are comparing several approaches to construct the initial conditions and proposed 'novel dimension reduction technique employing randomized singular value decomposition (SVD)' as a new aggregation technique. The manuscript is very well written and contributes to this research topic. The only concerns I have relate to the interpretation of results. A range of possible reasons for discrepancies in the apriori and a posteriori results are not considered even though these are mentioned in the Introduction. In addition, I think a direct comparison with the recent spatially resolved bottom up approach by Gerber et al. (2016) (see reference listed below) is needed. I have given some specific

suggestions for improvements below.

Title: 'optimal resolution': this term is mentioned in Introduction and M&M, but not in

Abstract/Conclusions. Perhaps it can be added to provide connections for reader.

Page 1 L. 19: Is a comma needed here 'global, monthly'?

L. 29: 'more' than? Please clarify.

L. 30: 'fertilizer': I assume authors are referring to inorganic fertilizer (as in main text) but $N_2O$ emissions are driven by all forms of N input (manure, crop residue, soil mineralization, wet and dry N deposition. Manure N addition could also be contributing to the seasonality.

L. 32: Please see my comments for this explanation below.

L. 33: 'aliasing': this term is not used elsewhere in text. It would be helpful to use terms consistently so connections between different sections of manuscript can be made.

Page 2 L. 9-10: '... attribution of the source to specific regions and sectors is hindered by the strong spatio-temporal variability in $N_2O$ emissions...': something seems amiss here. High spatial variability hinders source attribution to regions? Do you mean 'Sources ARE highly variable in space and time and this hinders top-down approaches because of... ( factors listed in the remaining text)?

L. 21: Manure N use also increased as shown by Davidson 2009 (cited here).

L. 25: indirect $N_2O$ emissions are also due to $NH_3$ volatilization; please include a reference to this.

L. 26: It is not just uncertainties in the indirect component that affect the global $N_2O$ budget. The non-linear response to N input rates (please see Gerber et al. 2016, Spatially explicit estimates of $N_2O$ emissions from croplands suggest climate mitigation opportunities from improved fertilizer management, GCB), uncertainties in manure

management estimates (e.g. manure deposited in pasture), and soil freeze/thaw effects are some examples of aspects that should be cited here.

L. 26: Omit 'a body of' as two studies do not seem to warrant this statement. In addition, the factors cited above (non-linear response, freeze/thaw, etc.) also point to over or under-estimates (depending on factor) and these should be mentioned here.

Page 3 L. 2: when fertilizer is applied is not necessarily the issue unless it coincides with favourable soil conditions. It may be useful to mention wet/dry cycles here (see Kim et al. 2012, Effects of soil rewetting and thawing on soil gas fluxes: a review of current literature and suggestions for future research, Biogeosci.) and how they interact with management of N input.

L. 4: I do not recall that this paper looked at duration of freeze-thaw cycles. From what I recall it is showing the global agric N2O budget could be underestimated by a certain amount due to these cycles. This seems to be the relevant aspect from that publication to cite here.

L. 32: I may have missed something but the airborne measurements were not used to directly assess optimized emissions, correct? (rather concentration profiles).

Page 4: L. 6: Why was this period chosen for simulation?

L. 15: Should mention that monthly values for N2O emissions from Edgar were used. Need to discuss here and/or later what drives the seasonal variation in this model and how/why it does not capture some of the seasonal variation discussed in Intro.

Page 9 L. 11: 'Remoteopt' used only observations from the remote sites, correct?

L. 21: 'remote sites': it would be helpful to list which ones are the remote sites, here and/or in table heading.

L. 26: Should mention evaluation was done for each hemisphere (as shown in table 2).

Page 10: L. 18-20: The sentence starting with 'However, because...' is hard to follow

and should be edited.

Page 11: L. 29: '....implying that the global annual a priori flux is too high.' How does this square with the arguments presented that some sources are underestimated in the bottom-up approaches? Please clarify.

Page 12: L. 17-18: It would be helpful to indicate the regions in Figure 7 where authors feel most confident of results and then discuss only these regions in detail.

L. 27: Please refer to Fig. 3 after 'Both the standard and SVD-based inversions call for a large increase (2-3×) in emissions from the US corn belt...'. Here and in the discussion that follows in is sometimes difficult to compare the a priori and a posteriori results. Perhaps plotting the difference (increase or decrease in comparison to the a priori map would help the reader to follow the presentation?

L. 30: I do not follow why the authors single out 'underrepresentation of the indirect N2O source associated with leaching and runoff from agricultural soils' as the likely reason for magnitude of upwards adjustment derived in this study. As suggested in the comments for introduction there are other factors that could be having an impact.

Pag 13: L. 1-2: Overestimation of natural emissions is used to explain the downward adjustment for western US and Canada. Could there possibly be other reasons? Gerber et al. 2016 show smaller fertilizer emission factors for these regions than usually used in inventories and this should also be considered here. A comparison with Gerber et al for the other regions should also be made (similar results seen for increases in emissions in southern China).

Page 15: L. 13-14: Can authors really state the reasons for disagreement? Please see comment above. Is it possible that regions in Western US and Canada have lower N2O emissions than the a priori model predicts due to lower fertilizer use and/or drier conditions (less use of irrigation?).

L. 18-19: I am not sure why 'Seasonality in our prior emissions is dominated by the

natural soil source.'. Wouldn't fertilizer related emissions also be seasonal?

L. 24: 'November – December peak, and a May – June minimum': this is difficult to see in the figure. Perhaps more detailed X-axis labels would help.

L. 25: Fix 'an a'.

L. 30: No need to use abbreviation (STE) as only used once. Page 16:

L. 5-6: It is possible that indirect emissions are the reason for discrepancies between measurements and model. Would this also be the case for other regions where the same model for the a priori emissions is used? Could the differences be due to freeze/thaw emissions or higher than expected direct N2O emissions due to high N application rates (in exponential part of non-linear response curve), which are not considered in the a priori emissions? Also, I am a bit confused by 'The fact that it is also one of the only sites located in an agricultural source region....'. Could such discrepancy only show up in places where measurements are done at an agricultural site? Are other agricultural source regions being missed because there are no monitoring sites close by?

L. 11-12: '... with the North American results exhibiting separate spring and summer peaks (plus a fall-winter enhancement in the SVD-based inversion)': I had difficulty seeing this in the figure. Perhaps better X-axis labels would help here as well.

Page 16: L. 28-29: '... which have been shown (Chen et al., 2016) to peak earlier (indirect emissions) and later (direct emissions) in the growing season': I am confused as to why the indirect emissions would peak earlier since they derive from N that is lost from the fertilizer application and then nitrified or denitrified in water ways (after leaching or run-off) and soils (after dry deposition). The earlier peak seems more consistent with emissions due to spring thaw. Conclusions: comments made above apply here as well.

Table 1: explain which sites are 'remote'.

Table 2: spell out SH, NH in heading.

Figure 2: Give time period (April 2010 to . . .) in caption and add 4/10 to X-axis labels. Use of letters in a more frequent interval may help reader find peaks/lows discussed in text.

Fig 3: Some pixels appear black on maps. Is that correct? It would be helpful to plot difference between two approaches instead of absolute amount so that areas of discrepancy can be identified more easily.

---

## Referee Comment (RC2) · Anonymous Referee #2 · 5 Oct 2017

This paper uses a multi-inversion hierarchy to derive top-down constraints on N2O emissions for 2011. The goal is to make a detailed evaluation of the 3 different methods and their impacts on inversion results. All methods are based on the adjoint of the GEOS-Chem chemical transport model, where 4D Var is considered the "standard" approach, as well as two alternative ways for aggregating the results, given that the existing observational network is insufficient to fully constrain N2O emissions at the gridscale level. The first approach uses the 4DVar method, but aggregated to the traditional 6 continents and 3 oceans. The more novel approach tested is a new SVD-based technique based on the "prior- preconditioned Hessian of the 4D-Var cost function."

[Figure]

An additional goal is to address the impact of initial condition uncertainties using 6 different approaches. This analysis is performed first and an optimal approach is selected for use in the evaluation of the 3 different inversion methods.

The paper is well written and logically organized. While some of the mathematics, particularly the SVD approach, are beyond my ability to evaluate, I found the results and discussion interesting and insightful. My main criticisms are, first, there seems to be a predisposition to claim the SVD results as the "best estimate of the true global flux." This conclusion is not clearly based on objective criteria. Other interpretations that might be more critical of SVD are not discussed, including the odd, spiky SVD results (e.g., in South America, Africa and the Tropical Oceans in Figure 7). Second, there is an unwarranted emphasis on the results of Chen et al. 2016, which are often presented as though they were primary results of the current study (see further comments below). However, these are minor criticisms of what is overall an impressive and interesting body of work. I recommend publication with some relatively minor revisions detailed below.

Specific comments

Abstract L31-32 "the inversions reveal a major emission underestimate in the US Corn Belt (which may extend to other intensive agricultural regions), likely from underrepresentation of indirect N2O emissions from leaching and runoff." Please clarify an underestimate relative to what? Also, the last part of this sentence is supported only on p12L30 with a reference to Chen et al. 2016. It is not supported by the current study and does not really belong in the abstract as a new primary finding.

As an aside, I will make a few comments about Chen et al. 2016, which is referenced multiple times (e.g, again on P17L27) as the source of the conclusion that the underestimate of indirect emissions is responsible for the underestimate of agricultural emissions in prior inventories. Realistically, I don't think the Chen et al. methodology is able to separate indirect and direct emissions. Their prior direct agricultural source

is based on EDGAR, which is at least somewhat reliable since it is computed using gridded N inputs from fertilizer, etc. multiplied by emission coefficients. In contrast, the indirect source is based on the CLM45-BGC nitrate leaching and runoff flux, which is unreliable and almost certainly wrong (see, e.g., Houlton et al., Nature Climate Change, 5, 398, 2015). The Chen methodology then assumes those 2 prior sources accurately represent the spatial and temporal distribution of direct and indirect $N_2O$ emissions, respectively. That methodology is fraught with uncertainty. Moreover, the fact that (as stated on p16L28) indirect emissions peak earlier than direct emissions is a red flag that something is wrong. This result doesn't make sense, given that indirect emissions, by IPCC definition, occur later and downstream/downwind of direct emissions.

P2L24 Crutzen et al., 2008; Davidson, 2009 are not really bottom-up emissions. They are based more on a top-down approach (in a global box model sense) of comparing the observed atmospheric $N_2O$ increase to the rate of external N inputs and anthropogenic N fixation.

P3L10-12 It seems somewhat over-critical to say previous aggregation has been informal and ad hoc. It's been based largely on geographical and political boundaries, i.e, North vs. South America, Pacific vs. Atlantic Ocean, etc., which are logical regions of interest.

P3L10-18 Exact totals are given for the ocean, GFED and EDGAR non-agricultural sources, but the Saikawa non-agricultural land and the EDGAR agricultural source are not specifically stated, yet these are the largest component sources. Please report them too. Two additional points are that the Saikawa source was based on a global model without cropland, such that it included a "non-agricultural" soil source from land such as the US Midwest where crops are grown. Also, the EDGAR v4.2 total is about 1.7 TgN/yr from industry, wastewater and energy. To bring up to the reported 2.3 TgN/yr, I wonder if the authors have included the EDGAR savanna, forest, grass and agricultural fire fluxes (of 0.84 Tg Nyr), which might be redundant with the GFED source? (Note: my numbers are from 2005 and thus may be slightly different from

2008.)

P4L29 Please state the time resolution of the inversion somewhere around here.

P5L19 Negative emission scaling factors may be appropriate for some oceanic regions, especially during seasonal cooling in regions where the biological source is small and thermal solubility-driven uptake may dominate the air-sea flux.

P5L27-29 Can we infer from this that the total observational uncertainty (which is also referred to as model-data mismatch uncertainty) is typically about 0.45 ppb? It would be useful to state this. It is interesting and unexpected that the observational uncertainty dominates the model representation uncertainty. At only 0.2 ppb, the model representation error seems substantially underestimated. Also, considering that the grid resolution is 4x5 degrees, how many grid boxes actually "surround" any given observation and what kind of heterogeneity is missed inside the actual box that contains the measurement?

P7 The SVD-method is complex to the point of being unfathomable for many readers (including me!), so we must take it on faith that the calculation is accurate. Given the lengthy form of equation 5, I am concerned that it would be easy for human errors to slip into the calculation. What assurances do we have that such errors will be detected?

P9-10 I found this section difficult to follow and did not emerge with a clear understanding of why certain initialization methods are better than others. I'm not sure what to suggest to help clarify, but one step might be to include some columns for the NH-SH gradient in Table 2 in addition to (or perhaps instead of) the separate NH and SH bias columns. Those are not really referred to in the text, while the "overly strong interhemispheric gradient" is mentioned on P9L26 but is not obvious in Table 2.

P11L8 The statement that SVD "appears to provide the best estimate of the true global flux" seems based on fairly limited and/or subjective criteria. Furthermore, it is not obviously true that SVD agrees best with HIPPO. In fact, it seems to agree worst from 30S-

30N. (This is attributed on P11 to the fact that "the spatial distribution (in the tropics) is particularly difficult to resolve," but this is not necessarily a satisfactory explanation.) Is the comparison to HIPPO based on subjective visual inspection or some more quantitative measure? Also, are we sure the HIPPO calibration scale is not systematically biased from the data used in the inversion, especially given the adjustments described in section 2.3?

P13 section 4.3.3. The results for Europe indicate a fairly dramatic reduction from the prior. Please state the total non-agricultural prior source in EDGARv4.2. How much of the total 1.70 Tg N prior does it comprise?

P14L31 The results of 3.35-3.48 are above the range found by Buitenhuis (2.4 +/- 0.8).

P17L20 It's perhaps notable here that the EDGAR industrial source has dropped by about a factor of 2 in version subsequent to v4.2 used here.

P17L23-24. I think the main issue is that the seasonality in the existing inventories used here is governed by natural soil emissions from a model without crops. The EDGAR agricultural source with no seasonality is then added. However, the hotspot of emission is in agricultural areas where the seasonality is influenced by spring fertilizer input. Thus the seasonality of the existing inventories was predictably wrong from the outset.

P31 Figure 3 caption. Do the bars show the median (as currently stated) or the mean of the 3 inversions? How meaningful is the median of just 3 values? Would it be better to just show all 3 results + prior, i.e., 4 bars per region?

P34. I'm not sure Figure 6 adds much value to the paper. Furthermore, why is the KCMP measurement of primary interest to the current study?

---

## Author Comment (AC1) · 17 Nov 2017

**Response to anonymous referee #1**

*The manuscript by Welles et al addresses an important need which is the comparison of top-down and bottom-up estimates of global N2O emissions. The authors are comparing several approaches to construct the initial conditions and proposed 'novel dimension reduction technique employing randomized singular value decomposition (SVD)' as a new aggregation technique. The manuscript is very well written and contributes to this research topic. The only concerns I have relate to the interpretation of results. A range of possible reasons for discrepancies in the a priori and a posteriori results are not considered even though these are mentioned in the Introduction. In addition, I think a direct comparison with the recent spatially resolved bottom up approach by Gerber et al. (2016) (see reference listed below) is needed. I have given some specific suggestions for improvements below.*

**We wish to thank the reviewer for their positive evaluation of our manuscript. Please find our responses to specific comments below, where the comment is in italics and our response is in bold.**

*Title: 'optimal resolution': this term is mentioned in Introduction and M&M, but not in Abstract/Conclusions. Perhaps it can be added to provide connections for reader.*

**Thank you for the suggestion. We have added the term "optimal resolution" in the abstract and in the conclusions.**

*Page 1 L. 19: Is a comma needed here 'global, monthly'?*

**We have deleted this comma, as well as another separating the same words in the conclusions.**

*L. 29: 'more' than? Please clarify.*

**The word "more" here referred to the a priori database. We have deleted the word and left "consistent with" to avoid confusion.**

*L. 30: 'fertilizer': I assume authors are referring to inorganic fertilizer (as in main text) but N2O emissions are driven by all forms of N input (manure, crop residue, soil mineralization, wet and dry N deposition. Manure addition could also be contributing to the seasonality.*

**Thank you for pointing this out. We have clarified this in the abstract as "spring fertilizer and manure application".**

*L. 32: Please see my comments for this explanation below.*

*L. 33: 'aliasing': this term is not used elsewhere in the text. It would be helpful to use term consistently so connections between different sections of manuscript can be made.*

**Thanks for the suggestion. We have changed the word "aliasing" to "biasing" to be more consistent with terms used elsewhere.**

*Page 2 L. 9-10: '… attribution of the source to specific regions and sectors is hindered by the strong spatio-temporal variability in N2O emissions…': something seems amiss here. High spatial variability hinders source attribution to regions? Do you mean 'Sources ARE highly variable in space and time and this hinders top-down approaches because of…(factors listed in remaining text)?*

**We have now reworded this sentence for greater clarity.**

*L. 21: Manure N use also increased as shown by Davidson 2009 (cited here).*

**We have now included manure N in this sentence.**

*L. 25: indirect N2O emissions are also due to NH3 volatilization; please include a reference to this.*

**Thank you for pointing out this omission. We now mention indirect emissions due to deposition of volatilized $NO_x$ and $NH_3$ here in the text.**

*L. 26: It is not just uncertainties in the indirect component that affect the global N2O budget. The non-linear response to N input rates (please see Gerber et al. 2016, Spatially explicit estimates of N2O emissions from croplands suggest climate mitigation opportunities from improved fertilizer management, GCB), uncertainties in manure management estimates (e.g. manure deposited in pasture), and soil freeze/thaw effects are some examples of aspects that should be cited here.*

**Indeed, we did not mean to imply here that the indirect emissions are the only source of uncertainty in the global agricultural N₂O budget. To that end, we now include a sentence here saying: "These sources are all subject to large uncertainties. For example…"" We already mention the nonlinear response to N input rates two sentences earlier. We have now also added a sentence specifically referencing Gerber et al. (2016) and the under or overestimate that can arise due to the non-linear response of emissions to fertilizer application. Freeze/thaw effects are already addressed in the following paragraph.**

*L. 26: Omit 'a body of' as two studies do not seem to warrant this statement. In addition, the factors cited above (non-linear response, freeze/thaw, etc.) also point to over or under-estimates (depending on factor) and these should be mentioned here.*

**We have deleted "a body of" from this sentence. Uncertainties in these other factors are now more explicitly addressed as described in our response to the previous comment.**

*Page 3 L. 2: when fertilizer is applied is not necessarily the issue unless it coincides with favourable soil conditions. It may be useful to mention wet/dry cycles here (see Kim et al. 2012, Effects of soil rewetting and thawing on soil gas fluxes: a review of current literature and suggestions for future research, Biogeosci.) and how they interact with management of N input.*

**This sentence did not say that fertilizer application timing was the main issue, just that microbial nitrification and denitrification depends on fertilizer application in general. In response to the reviewer's comment, we have now changed the wording to the following,**

and added a reference to Kim et al.: "Because microbial nitrification and denitrification, and the subsequent soil-atmosphere $N_2O$ flux, depend strongly on factors such as soil moisture, temperature, physical characteristics, and N availability (e.g., Potter et al., 1996; Bouwman, 1998; Kim et al., 2012; Bouwman et al., 2013; Butterbach-Bahl et al., 2013; Griffis et al., 2017), $N_2O$ emissions can exhibit major temporal and spatial variability."

*L. 4: I do not recall that this paper looked at duration of freeze-thaw cycles. From what I recall it is showing the global agric N2O budget could be underestimated by a certain amount due to these cycles. This seems to be the relevant aspect of that publication to cite here.*

This paper estimated a global $N_2O$ source during the non-growing season due to short-duration thaw events in seasonally frozen soils. Adding this to the current EDGAR direct source for these soils results in the 35-65% contribution cited. We have clarified in the text that this contribution is to the direct source, and added some text to mention what the total global agricultural underestimation could be: "For example, Wagner-Riddle et al. (2017) found that short-duration freeze-thaw cycles can account for 35-65% of the annual direct $N_2O$ emissions from seasonally frozen croplands, and that neglecting this contribution would lead to a 17-28% underestimate of the global $N_2O$ source (direct+indirect) from agricultural soils."

*L. 32: I may have missed something but the airborne measurements were not used to directly assess optimized emissions, correct?*

Correct. This is mentioned in Section 2.3 and in the caption of Fig. 4, but we have also added the word "independent" before "airborne measurements" here as a reminder that these were not used in the inversion.

*Page 4 L. 6: Why was this period chosen for simulation?*

$N_2O$ measurements started at the KCMP tall tower in April 2010, so our simulation period spans the first two years of observations at that site. We have added this explanation here.

*L. 15: Should mention that monthly values for N2O emissions from Edgar were used. Need to discuss here and/or later what drives the seasonal variation in this model and how/why it does not capture some of the seasonal variation discussed in Intro.*

We use annual emissions from EDGAR in our a priori. We have added the word "annual" here as a reminder, and have added some text when discussing the seasonality to note that the (monthly) natural soil source is driving the seasonality of emissions over land.

*Page 9 L. 11 'Remoteopt' used only observations from the remote sites, correct?*

Correct. We have clarified this here as follows: "Three involve interpolation of surface observations from the NOAA, AGAGE, CSIRO, EC, and NIWA networks for alternate time windows (MarZonal, AprZonal, AprKriging), two involve 4D-Var adjoint optimization of the initial mass field based on those same observations plus those from KCMP tall tower (AprOpt, FebOpt), and one involves optimization of the initial mass field based on observations from remote sites (RemoteOpt)."

*L. 21: 'remote sites': it would be helpful to list which ones are remote sites, here and/or in table heading.*

**Thanks for the suggestion. We have added a footnote to the table denoting which sites were considered remote.**

*L. 26: Should mention evaluation was done for each hemisphere (as shown in table 2).*

**We have added this clarification as follows: "Table 2 shows initial bias statistics with respect to all surface observations and by hemisphere for each initial condition treatment."**

*Page 10 L. 18-20: The sentence starting with 'However, because…' is hard to follow and should be edited.*

**We have broken this into two sentences to improve flow and clarity: "However, our a priori flux and lifetime are broadly consistent with independent observational constraints (Prather et al., 2012), whereas an annual $N_2O$ source of 20+ Tg N would yield a higher-than-observed atmospheric growth rate. A biased initial mass field is thus the more tenable explanation for the negative model:measurement residual trend."**

*Page 11 L. 29: '...implying that the global annual a priori flux is too high.' How does this square with the arguments presented that some sources are underestimated in bottom-up approaches? Please clarify.*

**We did not mean to imply that an underestimate of certain sources necessarily leads to an underestimate in the global source. We have added a clarification here as follows: "the global annual a priori flux (from all sources combined)".**

*Page 12 L. 17-18: It would be helpful to indicate the regions in Figure 7 where authors feel most confident of results and then discuss only these regions in detail.*

**Thank you for the suggestion, but we wish to retain comparisons to other studies for each region and feel the separate sections helps the reader quickly find results for a region of interest. For regions where the results are more uncertain due to low observational constraints we mention this explicitly in the corresponding section.**

*L. 27: Please refer to Fig. 3 after 'Both the standard and SVD-based inversions call for a large increase (2-3×) in emissions from the US corn belt…' Here and in the discussion that follows in is sometimes difficult to compare the a priori and a posteriori results. Perhaps plotting the difference (increase or decrease in comparison to the a priori map would help the reader to follow the presentation?*

**Thank you for the suggestion. We have now included a reference to Fig. 3 after that phrase in the text. We have also added maps of the a posteriori emission increment (a posteriori – a priori) in Fig. 3 to aid the reader in identifying areas of increase/decrease relative to the prior.**

*L. 30: I do not follow why the authors single out 'underrepresentation of the indirect N2O source associated with leaching and runoff from agricultural soils' as the likely reason for magnitude of*

*upwards adjustment derived in this study. As suggested in the comments for introduction there are other factors that could be having an impact.*

**The indirect source is the one most supported in the literature for this region, but other sources certainly could be contributing to the underestimate here. We have now added a mention of the potential impact of freeze-thaw and direct emissions here, as well as later in the conclusions.**

*Page 13 L. 1-2: Overestimation of natural emissions is used to explain the downward adjustment for western US and Canada. Could there possibly be other reasons? Gerber et al. 2016 show smaller fertilizer emission factors for these regions than usually used in inventories and this should also be considered here. A comparison with Gerber et al. for the other regions should also be made (similar results seen for increases in emissions in southern China).*

**Yes, it is certainly possible that direct agricultural emissions also contribute to the overestimate in western US and Canada. We have added a reference to Gerber et al. (2016) at the end of Section 4.3.1 to note the lower emission factors they found here. We also added a reference in Section 4.3.5 to support a potential underestimate of direct emissions in China when assuming a linear emission factor.**

*Page 15 L. 13-14: Can authors really state the reason for disagreement? Please see comment above. Is it possible that regions in western US and Canada have lower N2O emissions than the a priori model predicts due to lower fertilizer use and/or drier conditions (less use of irrigation?).*

**See our reply to the previous comment. We also clarify at the end that the underestimate from fertilized agricultural soils is specific to the US corn belt and possibly Asia where N input exceed crop demands.**

*L. 18-19: I am not sure why 'Seasonality in our prior emissions is dominated by the natural soil source.' Wouldn't fertilizer related emissions also be seasonal?*

**Yes, but the EDGARv4.2 emissions used here are annual so do not have any seasonality in the a priori emissions. We have added a reminder of this at the beginning of this sentence to clarify.**

*L. 24: 'November-December peak, and a May-June minimum': this is difficult to see in the figure. Perhaps are more detailed X-axis labels would help.*

**We have updated the x-axis here to be consistent with the updates made to Fig. 2: added 4/10 to the left-hand side of the axis, and slightly increased the interval of the axis labels.**

*L. 25: Fix 'an a'.*

**Thanks for catching this. We have fixed the error**

*L. 30; No need to use abbreviation (STE) as only used once.*

**We now spell out 'stratospheric-troposphere exchange' in place of the STE abbreviation here.**

*Page 16 L. 5-6: It is possible that indirect emissions are the reason for discrepancies between measurements and model. Would this also be the case for other regions where the same model for the a priori emissions is used? Could the differences be due to freeze/thaw emissions or higher than expected direct N2O emissions due to high N application rates (in exponential part of non-linear curve), which are not considered in the a priori emissions? Also, I am a bit confused by 'the fact that it is also one of the only sites located in an agricultural source region…' Could such discrepancy only show up in places where measurements are done at an agricultural site? Are other agricultural source regions being missed because there are no monitoring sites close by?*

**Here we simply meant to highlight the negative model-measurement bias at this site (as it is unique from other surface sites used in the inversion), and note that the sign of the bias here is consistent with earlier studies that link it to an underestimate of indirect emissions. The site is situated on drained lands, so the result may not be representative of other agricultural systems. We have rephrased as follows: "…inversion period. Located in an agricultural region composed mainly of drained lands, the low model bias is consistent with previous findings…"**

*L. 11-12: '...with the North American results exhibiting separate spring and summer peaks (plus a fall-winter enhancement in the SVD-based inversion)': I had difficulty seeing this in the figure. Perhaps better X-axis labels would help here as well.*

**We have now included x and y-axis labels on all panels in Fig. 7 to help the reader more quickly see when the peaks are occurring. The SVD-based peak referred to in the text occurs in October, so we now mention this explicitly in the text.**

*Page 16 L. 28-29: '...which have been shown (Chen et al., 2016) to peak earlier (indirect emissions) and later (direct emissions) in the growing season': I am confused as to why the indirect emissions would peak earlier since they derive from N that is lost from the fertilizer application and the nitrified or denitrified in water ways (after leaching or run-off) and soils (after dry deposition). The earlier peak seems more consistent with emissions due to spring thaw. Conclusions: comments made above apply here as well.*

**We note here that based on the IPCC definition: "Indirect pathways involve nitrogen that is removed from agricultural soils and animal waste management systems via volatilization, leaching, runoff, or harvest of crop biomass", so there is not an indication of seasonality here. Indirect emissions in the US Corn Belt are high in April-June when tile drainage and stream discharge peak. However, the reviewer is correct that spring thaw is also a possible contributor, and we have edited the text to reflect this. Additionally, many farmers in the US Corn Belt apply fertilizer in the fall, which would serve as a source of nitrogen to be released in the spring. As such, we have added the following sentence to the end of this section: "Fall fertilizer application is also common in the US Corn Belt—more than one third of corn farmers in Minnesota do their main N application during this time (Beirman**

**et al., 2012)—which could explain the October peak in the SVD-based results, and provide a source of nitrogen that would be released in the early spring thaw and subsequent runoff period." We also note in Section 4.3.1 the following: "However, other processes could also contribute, such as freeze-thaw emissions or direct emissions after spring fertilizer application. The timing of these processes, and that of peak stream flow, correspond to the dominant modes of ambient $N_2O$ variability observed in this region (Griffis et al., 2017)."**

*Table 1: explain which sites are 'remote'.*

**We have added a list of the remote sites as a footnote to the table.**

*Table 2: spell out SH, NH in heading*

**We have now spelled out Northern Hemisphere, Southern Hemisphere in the heading and caption of the table.**

*Figure 2: Give time period (April 2010 to…) in caption and add 4/10 to X-axis labels. Use of letters in a more frequent interval may help the reader find peaks/lows discussed in text.*

**Thank you for the suggestion. We have added the time period (April 2010 to April 2012) to the caption, added 4/10 to the left-hand side of the axis, and slightly increased the interval of the axis labels**

*Fig. 3: some pixels appear black on maps. Is that correct? It would be helpful to plot difference between two approaches instead of absolute amount so that areas of discrepancy can be identified more easily.*

**There are no black pixels in the emission maps in Fig. 3—perhaps this is just how the darkest red color appears in print? We have now added maps of the a posteriori emission increment (a posteriori – a priori) in Fig. 3 to aid the reader in identifying areas of increase/decrease relative to the prior.**

---

## Author Comment (AC2) · 17 Nov 2017

**Response to anonymous referee #2**

*This paper uses a multi-inversion hierarchy to derive top-down constraints on N2O emissions for 2011. The goal is to make a detailed evaluation of the 3 different methods and their impacts on inversion results. All methods are based on the adjoint of the GEOS-Chem chemical transport model, where 4D Var is considered the "standard" approach, as well as two alternative ways for aggregating the results, given that the existing observational network is insufficient to fully constrain N$_2$O emissions at the gridscale level. The first approach uses the 4DVar method, but aggregated to the traditional 6 continents and 3 oceans. The more novel approach tested is the new SVD-based technique based on the "prior- preconditioned Hessian of the 4D-Var cost function." An additional goal is to address the impact of initial condition uncertainties using 6 different approaches. This analysis is performed first and an optimal approach is selected for use in the evaluation of the 3 different inversion methods.*

*The paper is well written and logically organized. While some of the mathematics, particularly the SVD approach, are beyond my ability to evaluate, I found the results and discussion interesting and insightful. My main criticisms are, first, there seems to be a predisposition to claim the SVD results as the "best estimate of the true global flux." This conclusion is not clearly based on objective criteria. Other interpretations that might be more critical of SVD are not discussed, including the odd, spiky SVD results (e.g., in South America, Africa and the Tropical Oceans in Figure 7). Second, there is an unwarranted emphasis on the results of Chen et al. 2016, which are often presented as though they were primary results of the current study (see further comments below). However, these are minor criticisms of what is overall an impressive and interesting body of work. I recommend publication with some relatively minor revisions detailed below.*

**We wish to thank the reviewer for their positive evaluation of our manuscript. Please find our responses to specific comments below, where the comment is in italics and our response is in bold.**

*Abstract L31-32 "the inversions reveal a major emission underestimate in the US Corn Belt (which may extend to other regions), likely from underrepresentation of indirect N2O emissions from leaching and runoff. Please clarify an underestimate relative to what? Also, the last part of this sentence is supported only on p12L30 with a reference to Chen et al. 2016. It is not supported by the current study and does not really belong in the abstract as a primary new finding.*

**We have deleted the reference to leaching and runoff here, and clarified that the underestimate is in the prior bottom-up inventory used.**

*As an aside, I will make a few comments about Chen et al. 2016, which is referenced multiple times (e.g., again on P17L27) as the source of the conclusion that the underestimate of indirect emissions is responsible for the underestimate of agricultural emissions in prior inventories. Realistically, I don't think the Chen et al. methodology is able to separate indirect and direct*

*emissions. Their prior direct agricultural source is based on EDGAR, which is at least somewhat reliable since it is computed using gridded N inputs from fertilizer, etc. multiplied by emission coefficients. In contrast, the indirect source is based on the CLM45-BGC nitrate leaching and runoff flux, which is unreliable and almost certainly wrong (see, e.g., Houlton et al., Nature Climate Change, 5, 398, 2015). The Chen methodology then assumes those 2 prior sources accurately represent the spatial and temporal distribution of direct and indirect N2O emissions, respectively. That methodology is fraught with uncertainty. Moreover, the fact that (as stated on p16L28) indirect emissions peak earlier than direct emissions is a red flag that something is wrong. This result doesn't make sense, given that indirect emissions, by IPCC definition, occur later and downstream/downwind of direct emissions.*

**We would like to clarify a few points in the above comments on the Chen et al. (2016) paper. First, the Houlton et al. (2015) paper cited by the reviewer used the CLM-CN coupled model, which has a different solution for nitrate leaching and runoff than the CLM45-BGC model used in the Chen et al. (2016) paper. Second, the Chen et al. (2016) methodology does not assume the temporal distribution of a priori emissions is correct, as they solve for monthly fluxes. Third, a more recent paper (Griffis et al., PNAS, 2017), obtains very similar seasonality using different a priori emissions, which supports the Chen findings. However, Chen et al. (2016) is not being reviewed here and so we focus the rest of our response on issues relevant to our manuscript.**

**Based on the IPCC definition: "Indirect pathways involve nitrogen that is removed from agricultural soils and animal waste management systems via volatilization, leaching, runoff, or harvest of crop biomass", so there is not an indication of seasonality here. Indirect emissions in the US Corn Belt are high in April-June when tile drainage and stream discharge peak. Additionally, many farmers in the US Corn Belt apply fertilizer in the fall, which would serve as a source of nitrogen to be released in the spring. As such, we have added the following sentence to the end of this section: "Fall fertilizer application is also common in the US Corn Belt—more than one third of corn farmers in Minnesota do their main N application during this time (Beirman et al., 2012)—which could explain the October peak in the SVD-based results, and provide a source of nitrogen that would be released in the early spring thaw and subsequent runoff period." We also added the following sentences in Section 4.3.1 to indicate that different processes (beyond just leaching and runoff) could be driving the overall underestimate of emissions in this region: "However, other processes could also contribute, such as freeze-thaw emissions or direct emissions after spring fertilizer application. The timing of these processes, and that of peak stream flow, correspond to the dominant modes of ambient N₂O variability observed in this region (Griffis et al., 2017)."**

*P2L24 Crutzen et al., 2008; Davidson, 2009 are not really bottom-up emissions. They are based more on a top-down approach (in a global box model sense) of comparing the observed atmospheric N2O increase to the rate of external N inputs and anthropogenic N fixation.*

**Thank you for the clarification. We have deleted the term "Bottom-up" from the beginning of this sentence.**

*P3L10-12 It seems somewhat overcritical to say previous aggregation has been informal and ad hoc. It's been based largely on geographical and political boundaries, i.e. North vs. South America, Pacific vs. Atlantic Ocean, etc., which are logical regions of interest.*

**Good point. We have replaced the phrase "in an informal ad-hoc way" to "based on physical or political boundaries".**

*P3L10-18 Exact totals are given for the ocean, GFED and EDGAR non-agricultural sources, but the Saikawa non-agricultural source and the EDGAR agricultural source are not specifically stated, yet these are the largest component sources. Please report them too. Two additional points are that the Saikawa source was based on a global model without cropland, such that it included a "non-agricultural" soil source from land such as the US Midwest where crops are grown. Also, the EDGARv4.2 total is about 1.7 TgN/yr from industry, waste water and energy. To bring up the reported 2.3 TgN/yr, I wonder if the authors have included the EDGAR savanna, forest, grass, and agricultural fire fluxes (of 0.84 Tg N/yr), which might be redundant with the GFED source? (Note: my numbers are from 2005 and thus may be slightly different from 2008.)*

**Thank you for catching this. We did not include the EDGAR fire sources in our 2.3 Tg N yr$^{-1}$, but accidentally included indirect emission from $NO_x$ and $NH_3$ deposition (~0.4 Tg N) and manure management (~0.2 Tg N) in this total rather than in the reported agricultural source total. Thus, the EDGARv4.2 total for 2008 is about 1.7 Tg N yr$^{-1}$ as you noted. We now report specific totals for the Saikawa source (7.5 Tg N) and EDGAR agricultural soil direct+indirect (3.5 Tg N yr$^{-1}$) and manure management sources separately.**

*P4L29 Please state the time resolution of the inversion somewhere around here.*

**We have added "monthly" to the first line of this paragraph.**

*P5L19 Negative emission scaling factors may be appropriate for some oceanic regions, especially during seasonal cooling in regions where the biological source is small and thermal solubility-driven uptake may dominate the air-sea flux.*

**We do have negative fluxes where uptake dominates the air-sea flux in our a priori oceanic emissions. Our inversion approach does not require positive fluxes, but it does assume that the sign of the a priori flux is correct in each grid square. We now mention this explicitly in the text here.**

*P5L27-29 Can we infer from this that the total observational uncertainty (which is also referred to as model-data mismatch uncertainty) is typically about 0.45 ppb? It would be useful to state this. It is interesting and unexpected that the observational uncertainty dominates the model representation uncertainty. At only 0.2 ppb, the model representation error seems substantially underestimated. Also, considering that the grid resolution is 4x5 degrees, how many grid boxes actually "surround" any given observation and what kind of heterogeneity is missed inside the actual box that contains the measurement?*

**Correct, this would correspond to a mean observational error of ~0.45 ppb, which we now mention in the text. We also note that values extend up to ~4 ppb. Given the coarse**

**horizontal resolution, we could be underestimating the representation error for near-source observations. However, we have since run a test standard inversion with tripled observational error and get very similar results (global flux of 17.8 Tg N).**

*P7 The SVD-method is complex to the point of being unfathomable for many readers (including me!), so we must take it on faith that the calculation is accurate. Given the lengthy form of equation 5, I am concerned that it would be easy for human errors to slip into the calculation. What assurances do we have that such errors will be detected?*

**Three of the co-authors have rechecked the equations for accuracy, and no errors have been detected.**

*P9-10 I found this section difficult to follow and did not emerge with a clear understanding of why certain initialization methods are better than others. I'm not sure what to suggest to help clarify, but one step might be to include some columns for the NH-SH gradient in Table 2 in addition to (or perhaps instead of) the separate NH and SH bias columns. Those are not really referred to in the text, while the "overly strong interhemispheric gradient" is mentioned on P9L26 but is not obvious in Table 2.*

**We have tried to clarify here that an overly strong interhemispheric gradient is indicated by the fact that the model has a high bias in the Northern Hemisphere and a low bias in the Southern Hemisphere. We have also removed the subsequent reference to the interhemispheric gradient and replaced it with the following sentence: "The interpolation methods without subsequent spinup (AprZonal, AprKriging) perform better in terms of initial model:measurement bias – in the global mean and in each individual hemisphere."**

*P11L18 The statement that SVD "appears to provide the best estimate of the true global flux" seems based on fairly limited and/or subjective criteria. Furthermore, it is not obviously true that SVD agrees best with HIPPO. In fact, it seems to agree worst from 30S-30N. (This is attributed on P11 to the fact that "the spatial distribution (in the tropics) is particularly difficult to resolve," but this is not necessarily a satisfactory explanation.) Is the comparison to HIPPO based on subjective visual inspection or some more quantitative measure? Also, are we sure the HIPPO calibration scale is not systematically biased from the data used in the inversion, especially given the adjustments described in section 2.3?*

**We have now deleted the claim that the SVD-based inversion provides the best estimate of the true global flux. We also outline more specifically where/when the agreement with HIPPO is improved. The sentence now reads: "It also gives a better comparison to HIPPO IV and V measurements in the southern extratropics and to HIPPO V in the northern extratropics (see below)." We have also edited the last two sentences of Section 4.1 to read "The lower global flux obtained with the SVD-based approach (Fig. 3 and Table 3) is thus the reason for this correction, implying that the global annual a priori flux (from all sources combined) may be too high. We note that a slight low bias does emerge in the tropics in the SVD-based approach, where observational constraints are low." We have also edited the language in the conclusions to be consistent with this. As for the calibration,**

**we have adjusted the HIPPO QCLS data based on concurrent flask observations, which are on the NOAA scale. We now mention this at the end of Section 2.3.**

*P13 section 4.3.3 The results for Europe indicate a fairly dramatic reduction from the prior. Please state the total non-agricultural prior source in EDGARv4.2. How much of the total 1.70 Tg N prior does it comprise?*

**We had already included the contribution of EDGARv4.2 non-agricultural sources to the total in Europe (~40%) in Section 4.3.3. However, since we accidentally lumped manure management and indirect emissions from $NO_x$ and $NH_3$ deposition in that total, we have revised this number. The total European non-agricultural source in EDGARv4.2 is about 0.5 Tg N, which is about 30% of the total prior emissions here. The resulting relative a posteriori adjustments for soil and non-agricultural sources, when integrated over Europe, are thus comparable in magnitude, and we have edited the text to reflect that.**

*P14L31 The results of 3.35-3.48 are above the range found by Buitenhuis (2.4 +/- 0.8).*

**Correct, we already note at the end of this sentence that our optimized oceanic fluxes are higher than that found by Buitenhuis. However, they are closer to that estimate than the results of Thompson et al. (2014), which is what we meant by "more consistent with". We have now clarified this in the text.**

*P17L20 It's perhaps notable here that the EDGAR industrial source has dropped by about a factor of 2 in version subsequent to v4.2 used here.*

**Thank you for mentioning this. We have added a note mentioning this at the end of this bullet point in the text.**

*P17L23-24 I think the main issue here is that the seasonality in the existing inventories used here is governed by natural soil emissions from a model without crops. The EDGAR agricultural source with no seasonality is then added. However, the hotspot of emission is in agricultural areas where the seasonality is influenced by spring fertilizer input. Thus the seasonality of existing inventories is predictably wrong from the outset.*

**The reviewer is correct that we should expect some degree of seasonal bias given that annual EDGAR fluxes were used a priori. We have now deleted the phrase "than our current inventories suggest" from this sentence, and emphasize that the optimized seasonality is consistent with other studies.**

*P31 Figure 3 caption. Do the bars show the median (as currently stated) or the mean of the 3 inversions? How meaningful is the median of just 3 values? Would it be better to just show all 3 results + prior, i.e. 4 bars per region?*

**The bars do show the median of the a posteriori values as stated—the median was chosen so it is easy to infer all three values (min, median, and max) from the figure. We previously tried plotting 4 bars per region but found the plot too busy, so prefer to keep the two bars per region, but we have thickened the a posteriori error bars to more easily see the range.**

*P34. I'm not sure Figure 6 adds much value to the paper. Furthermore, why is the KCMP measurement of primary interest to the current study?*

**KCMP is of primary interest to our study given that i) it is the only site with a near-persistent model underestimate, ii) it is in an agricultural region comprised of drained lands (now mentioned at the end of Section 4.4.1), and iii) the emission processes for this ecosystem type are not well represented in current emission inventories, and have been linked to underestimated indirect $N_2O$ emissions associated with leaching and runoff. As for the value of Fig. 6, we find it helpful to show the seasonal model biases in $N_2O$ mixing ratio that exist before showing the a priori and a posteriori seasonal fluxes in Fig. 7.**